# Solving Inverse Physics Problems with Score Matching

Benjamin Holzschuh [1,2][*]    Simona Vegetti [2]    Nils Thuerey [1]

[1]Technical University of Munich, 85748 Garching, Germany
[2]Max Planck Institute for Astrophysics, 85748 Garching, Germany

## Abstract

We propose to solve inverse problems involving the temporal evolution of physics systems by leveraging recent advances from diffusion models. Our method moves the system's current state backward in time step by step by combining an approximate inverse physics simulator and a learned correction function. A central insight of our work is that training the learned correction with a single-step loss is equivalent to a score matching objective, while recursively predicting longer parts of the trajectory during training relates to maximum likelihood training of a corresponding probability flow. We highlight the advantages of our algorithm compared to standard denoising score matching and implicit score matching, as well as fully learned baselines for a wide range of inverse physics problems. The resulting inverse solver has excellent accuracy and temporal stability and, in contrast to other learned inverse solvers, allows for sampling the posterior of the solutions. Code and experiments are available at `https://github.com/tum-pbs/SMDP`.

## 1   Introduction

Many physical systems are time-reversible on a microscopic scale. For example, a continuous material can be represented by a collection of interacting particles [Gur82; BLL02] based on which we can predict future states. We can also compute earlier states, meaning we can evolve the simulation backward in time [Mar+96]. When taking a macroscopic perspective, we only know average quantities within specific regions [Far93], which constitutes a loss of information, and as a consequence, time is no longer reversible. In the following, we target inverse problems to reconstruct the distribution of initial macroscopic states for a given end state. This problem is genuinely tough [ZDG96; Góm+18; Del+18; LP22], and existing methods lack tractable approaches to represent and sample the distribution of states.

Our method builds on recent advances from the field of diffusion-based approaches [Soh+15; HJA20; Son+21b]: Data samples $\mathbf{x} \in \mathbb{R}^D$ are gradually corrupted into Gaussian white noise via a stochastic differential equation (SDE) $d\mathbf{x} = f(\mathbf{x}, t)dt + g(t)dW$, where the deterministic component of the SDE $f : \mathbb{R}^D \times \mathbb{R}_{\geq 0} \to \mathbb{R}^D$ is called *drift* and the coefficient of the $D$-dimensional Brownian motion $W$ denoted by $g : \mathbb{R}_{\geq 0} \to \mathbb{R}_{\geq 0}$ is called *diffusion*. If the **score** $\nabla_{\mathbf{x}} \log p_t(\mathbf{x})$ of the data distribution $p_t(\mathbf{x})$ of corrupted samples at time $t$ is known, then the dynamics of the SDE can be reversed in time, allowing for the sampling of data from noise. Diffusion models are trained to approximate the score with a neural network $s_\theta$, which can then be used as a plug-in estimate for the reverse-time SDE.

However, in our physics-based approach, we consider an SDE that describes the physics system as $d\mathbf{x} = \mathcal{P}(\mathbf{x})dt + g(t)dW$, where $\mathcal{P} : \mathbb{R}^D \to \mathbb{R}^D$ is a physics simulator that replaces the drift term of diffusion models. Instead of transforming the data distribution to noise, we transform a simulation state at $t = 0$ to a simulation state at $t = T$ with Gaussian noise as a perturbation. Based on a given

---

[*]Correspondence to: `benjamin.holzschuh@tum.de`

37th Conference on Neural Information Processing Systems (NeurIPS 2023).

end state of the system at $t = T$, we predict a previous state by taking a small time step backward in time and repeating this multiple times. Similar to the reverse-time SDE of diffusion models, the prediction of the previous state depends on an approximate inverse of the physics simulator, a learned update $s_\theta$, and a small Gaussian perturbation.

The training of $s_\theta$ is similar to learned correction approaches for numerical simulations [Um+20; Koc+21; LCT22]: The network $s_\theta$ learns corrections to simulation states that evolve over time according to a physics simulator so that the corrected trajectory matches a target trajectory. In our method, we target learning corrections for the "reverse" simulation. Training can either be based on single simulation steps, which only predict a single previous state, or be extended to rollouts for multiple steps. The latter requires the differentiability of the inverse physics step [Thu+21].

Importantly, we show that under mild conditions, learning $s_\theta$ is equivalent to matching the score $\nabla_{\mathbf{x}} \log p_t(\mathbf{x})$ of the training data set distribution at time $t$. Therefore, sampling from the reverse-time SDE of our physical system SDE constitutes a theoretically justified method to sample from the correct posterior.

While the training with single steps directly minimizes a score matching objective, we show that the extension to multiple steps corresponds to maximum likelihood training of a related neural ordinary differential equation (ODE). Considering multiple steps is important for the stability of the produced trajectories. Feedback from physics and neural network interactions at training time leads to more robust results.

In contrast to previous diffusion methods, we include domain knowledge about the physical process in the form of an approximate inverse simulator that replaces the drift term of diffusion models [Son+21b; ZC21]. In practice, the learned component $s_\theta$ corrects any errors that occur due to numerical issues, e.g., the time discretization, and breaks ambiguities due to a loss of information in the simulation over time.

Figure 1 gives an overview of our method. Our central aim is to show that the combination of diffusion-based techniques and differentiable simulations has merit for solving inverse problems and to provide a theoretical foundation for combining PDEs and diffusion modeling. In the following, we refer to methods using this combination as *score matching via differentiable physics* (SMDP). The main contributions of our work are: (1) We introduce a reverse physics simulation step into diffusion models to develop a probabilistic framework for solving inverse problems. (2) We provide the theoretical foundation that this combination yields learned corrections representing the score of the underlying data distribution. (3) We highlight the effectiveness of SMDP with a set of challenging inverse problems and show the superior performance of SMDP compared to a range of stochastic and deterministic baselines.

## 2  Related Work

**Diffusion models and generative modeling with SDEs**  Diffusion models [Soh+15; HJA20] have been considered for a wide range of generative applications, most notably for image [DN21], video [Ho+22; Höp+22; YSM22], audio synthesis [Che+21], uncertainty quantification [CSY22; Chu+22; Son+22; KVE21; Ram+20], and as autoregressive PDE-solvers [KCT23]. However, most approaches either focus on the denoising objective common for tasks involving natural images or the synthesis process of solutions does not directly consider the underlying physics. Models based on Langevin dynamics [Vin11; SE19] or discrete Markov chains [Soh+15; HJA20] can be unified in a time-continuous framework using SDEs [Son+21b]. Synthesizing data by sampling from neural SDEs has been considered by, e.g., [Kid+21; Son+21b]. Contrary to existing approaches, the drift in our method is an actual physics step, and the underlying SDE does not transform a data distribution to noise but models the temporal evolution of a physics system with stochastic perturbations.

**Methods for solving inverse problems for (stochastic) PDEs**  Differentiable solvers for physics dynamics can be used to optimize solutions of inverse problems with gradient-based methods by backpropagating gradients through the solver steps [Thu+21]. Learning-based approaches directly learn solution operators for PDEs and stochastic PDEs, i.e., mappings between spaces of functions, such as Fourier neural operators [Li+21], DeepONets [Lu+21], or generalizations thereof that include stochastic forcing for stochastic PDEs, e.g., neural stochastic PDEs [SLG22]. Recently, there have been several approaches that leverage the learned scores from diffusion models as data-driven

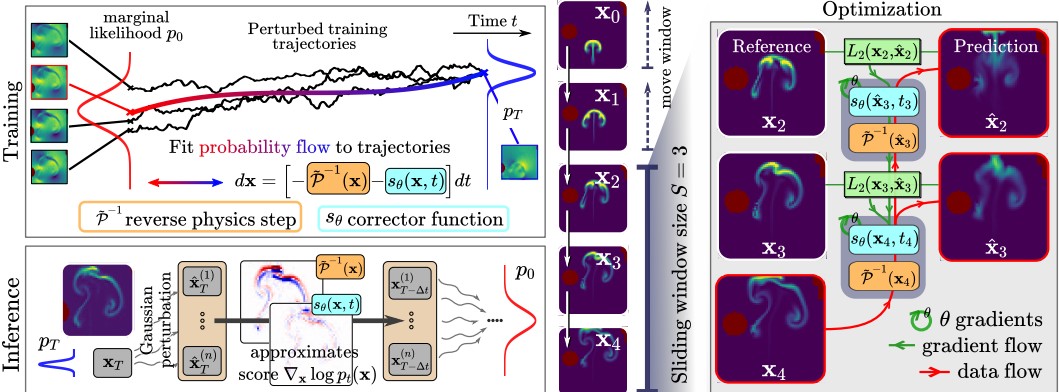

(a) Training and inference overview.  (b) Sliding window and optimizing $s_\theta$.

Figure 1: Overview of our method. For training, we fit a neural ODE, the probability flow, to the set of perturbed training trajectories (a, top). The probability flow is comprised of a reverse physics simulator $\tilde{\mathcal{P}}^{-1}$ that is an approximate inverse of the forward solver $\mathcal{P}$ as well as a correction function $s_\theta$. In many cases, we can obtain $\tilde{\mathcal{P}}^{-1}$ from $\mathcal{P}$ by using a negative step size $\Delta t$ or by learning a surrogate model from data. For inference, we simulate the system backward in time from $\mathbf{x}_T$ to $\mathbf{x}_0$ by combining $\tilde{\mathcal{P}}^{-1}$, the trained $s_\theta$ and Gaussian noise in each step (a, bottom). For optimizing $s_\theta$, our approach moves a sliding window of size $S$ along the training trajectories and reconstructs the current window (b). Gradients for $\theta$ are accumulated and backpropagated through all prediction steps.

regularizations for linear inverse problems [Ram+20; Son+22; KVE21; CSY22; Chu+22] and general noisy inverse problems [Chu+23]. Our method can be applied to general non-linear inverse physics problems with temporal evolution, and we do not require to backpropagate gradients through all solver steps during inference. This makes inference significantly faster and more stable.

**Learned corrections for numerical errors**   Numerical simulations benefit greatly from machine learning models [Tom+17; Mor+18; Pfa+20; Li+21]. By integrating a neural network into differential equation solvers, it is possible to learn to reduce numerical errors [Um+20; Koc+21; BWW22] or guide the simulation towards a desired target state [HTK20; Li+22]. The optimization of $s_\theta$ with the 1-step and multi-step loss we propose in section 3.1 is conceptually similar to learned correction approaches. However, this method has, to our knowledge, not been applied to correcting the "reverse" simulation and solving inverse problems.

**Maximum likelihood training and continuous normalizing flows**   Continuous normalizing flows (CNFs) are invertible generative models based on neural ODEs [Che+18a; KPB20; Pap+21], which are similar to our proposed physics-based neural ODE. The evolution of the marginal probability density of the SDE underlying the physics system is described by Kolmogorov's forward equation [Øks03], and there is a corresponding probability flow ODE [MRO20; Son+21b]. When the score is represented by $s_\theta$, this constitutes a CNF and can typically be trained with standard methods [Che+18b] and maximum likelihood training [Son+21a]. Huang et al. [HLC21] show that minimizing the score-matching loss is equivalent to maximizing a lower bound of the likelihood obtained by sampling from the reverse-time SDE. A recent variant combines score matching with CNFs [ZC21] and employs joint learning of drift and corresponding score for generative modeling. To the best of our knowledge, training with rollouts of multiple steps and its relation to maximum likelihood training have not been considered so far.

## 3   Method Overview

**Problem formulation**   Let $(\Omega, \mathcal{F}, P)$ be a probability space and $W(t) = (W_1(t), ..., W_D(t))^T$ be a $D$-dimensional Brownian motion. Moreover, let $\mathbf{x}_0$ be a $\mathcal{F}_0$-measurable $\mathbb{R}^D$-valued random variable that is distributed as $p_0$ and represents the initial simulation state. We consider the time evolution of

the physical system for $0 \leq t \leq T$ modeled by the stochastic differential equation (SDE)

$$d\mathbf{x} = \mathcal{P}(\mathbf{x})dt + g(t)dW \tag{1}$$

with initial value $\mathbf{x}_0$ and Borel measurable drift $\mathcal{P} : \mathbb{R}^D \to \mathbb{R}^D$ and diffusion $g : [0, T] \to \mathbb{R}_{\geq 0}$. This SDE transforms the marginal distribution $p_0$ of initial states at time 0 to the marginal distribution $p_T$ of end states at time $T$. We include additional assumptions in appendix A.

Moreover, we assume that we have sampled $N$ trajectories of length $M$ from the above SDE with a fixed time discretization $0 \leq t_0 < t_1 < ... < t_M \leq T$ for the interval $[0, T]$ and collected them in a training data set $\{(\mathbf{x}_{t_m}^{(n)})_{i=0}^M\}_{n=0}^N$. For simplicity, we assume that all time steps are equally spaced, i.e., $t_{m+1} - t_m := \Delta t$. Moreover, in the following we use the notation $\mathbf{x}_{\ell:m}$ for $0 \leq \ell < m \leq M$ to refer to the trajectory $(\mathbf{x}_{t_\ell}, \mathbf{x}_{t_{\ell+1}}, ..., \mathbf{x}_{t_m})$. We include additional assumptions in appendix A.

Our goal is to infer an initial state $\mathbf{x}_0$ given a simulation end state $\mathbf{x}_M$, i.e., we want to sample from the distribution $p_0(\,\cdot\,|\,\mathbf{x}_M)$, or obtain a maximum likelihood solution.

## 3.1 Learned Corrections for Reverse Simulation

In the following, we furthermore assume that we have access to a reverse physics simulator $\tilde{\mathcal{P}}^{-1} : \mathbb{R}^D \to \mathbb{R}^D$, which moves the simulation state backward in time and is an approximate inverse of the forward simulator $\mathcal{P}$ [HKT22]. In our experiments, we either obtain the reverse physics simulator from the forward simulator by using a negative step size $\Delta t$ or by learning a surrogate model from the training data. We train a neural network $s_\theta(\mathbf{x}, t)$ parameterized by $\theta$ such that

$$\mathbf{x}_m \approx \mathbf{x}_{m+1} + \Delta t \left[\tilde{\mathcal{P}}^{-1}(\mathbf{x}_{m+1}) + s_\theta(\mathbf{x}_{m+1}, t_{m+1})\right]. \tag{2}$$

In this equation, the term $s_\theta(\mathbf{x}_{m+1}, t_{m+1})$ corrects approximation errors and resolves uncertainties from the Gaussian perturbation $g(t)dW$. Below, we explain our proposed 1-step training loss and its multi-step extension before connecting this formulation to diffusion models in the next section.

**1-step loss**  For a pair of adjacent samples $(\mathbf{x}_m, \mathbf{x}_{m+1})$ on a data trajectory, the 1-step loss for optimizing $s_\theta$ is the $L_2$ distance between $\mathbf{x}_m$ and the prediction via (2). For the entire training data set, the loss becomes

$$\mathcal{L}_{\text{single}}(\theta) := \frac{1}{M} \mathbb{E}_{\mathbf{x}_{0:M}} \left[\sum_{m=0}^{M-1} \left[\left\|\mathbf{x}_m - \mathbf{x}_{m+1} - \Delta t \left[\tilde{\mathcal{P}}^{-1}(\mathbf{x}_{m+1}) + s_\theta(\mathbf{x}_{m+1}, t_{m+1})\right]\right\|_2^2\right]\right]. \tag{3}$$

Computing the expectation can be thought of as moving a window of size two from the beginning of each trajectory until the end and averaging the losses for individual pairs of adjacent points.

**Multi-step loss**  As each simulation state depends only on its previous state, the 1-step loss should be sufficient for training $s_\theta$. However, in practice, approaches that consider a loss based on predicting longer parts of the trajectories are more successful for training learned corrections [Bar+19; Um+20; Koc+21]. For that purpose, we define a hyperparameter $S$, called sliding window size, and write $\mathbf{x}_{i:i+S} \in \mathbb{R}^{S \times D}$ to denote the trajectory starting at $\mathbf{x}_i$ that is comprised of $\mathbf{x}_i$ and the following $S - 1$ states. Then, we define the multi-step loss as

$$\mathcal{L}_{\text{multi}}(\theta) := \frac{1}{M} \mathbb{E}_{\mathbf{x}_{0:M}} \left[\sum_{m=0}^{M-S+1} \left[\|\mathbf{x}_{m:m+S-1} - \hat{\mathbf{x}}_{m:m+S-1}\|_2^2\right]\right], \tag{4}$$

where $\hat{\mathbf{x}}_{i:i+S-1}$ is the predicted trajectory that is defined recursively by

$$\hat{\mathbf{x}}_{i+S} = \mathbf{x}_{i+S} \quad \text{and} \quad \hat{\mathbf{x}}_{i+S-1-j} = \hat{\mathbf{x}}_{i+S-j} + \Delta t \left[\tilde{\mathcal{P}}^{-1}(\hat{\mathbf{x}}_{i+S-j}) + s_\theta(\hat{\mathbf{x}}_{i+S-j}, t_{i+S-j})\right]. \tag{5}$$

## 3.2 Learning the Score

**Denoising score matching**  Given a distribution of states $p_t$ for $0 < t < T$, we follow [Son+21b] and consider the score matching objective

$$\mathcal{J}_{\text{SM}}(\theta) := \frac{1}{2} \int_0^T \mathbb{E}_{\mathbf{x} \sim p_t} \left[\|s_\theta(\mathbf{x}, t) - \nabla_\mathbf{x} \log p_t(\mathbf{x})\|_2^2\right] dt, \tag{6}$$

i.e., the network $s_\theta$ is trained to approximate the score $\nabla_\mathbf{x} \log p_t(\mathbf{x})$. In denoising score matching [Vin11; Soh+15; HJA20], the distributions $p_t$ are implicitly defined by a noising process that is given by the forward SDE $d\mathbf{x} = f(\mathbf{x}, t)dt + g(t)dW$, where $W$ is the standard Brownian motion. The function $f : \mathbb{R}^D \times \mathbb{R}_{\geq 0} \to \mathbb{R}^D$ is called *drift*, and $g : \mathbb{R}_{\geq 0} \to \mathbb{R}_{\geq 0}$ is called *diffusion*. The process transforms the training data distribution $p_0$ to a noise distribution that is approximately Gaussian $p_T$. For affine functions $f$ and $g$, the transition probabilities are available analytically, which allows for efficient training of $s_\theta$. It can be shown that under mild conditions, for the forward SDE, there is a corresponding **reverse-time SDE** $d\mathbf{x} = [f(\mathbf{x}, t) - g(t)^2 \nabla_\mathbf{x} \log p_t(\mathbf{x})]dt + g(t)d\tilde{W}$ [And82]. In particular, this means that given a marginal distribution of states $p_T$, which is approximately Gaussian, we can sample from $p_T$ and simulate paths of the reverse-time SDE to obtain samples from the data distribution $p_0$.

**Score matching, probability flow ODE and 1-step training**  There is a deterministic ODE [Son+21b], called probability flow ODE, which yields the same transformation of marginal probabilities from $p_T$ to $p_0$ as the reverse-time SDE. For the physics-based SDE (1), it is given by

$$d\mathbf{x} = \left[ \mathcal{P}(\mathbf{x}) - \frac{1}{2}g(t)^2 \nabla_\mathbf{x} \log p_t(\mathbf{x}) \right] dt. \tag{7}$$

For $\Delta t \to 0$, we can rewrite the update rule (2) of the training as

$$d\mathbf{x} = \left[ -\tilde{\mathcal{P}}^{-1}(\mathbf{x}) - s_\theta(\mathbf{x}, t) \right] dt. \tag{8}$$

Therefore, we can identify $\tilde{\mathcal{P}}^{-1}(\mathbf{x})$ with $-\mathcal{P}(\mathbf{x})$ and $s_\theta(\mathbf{x}, t)$ with $\frac{1}{2}g(t)\nabla_\mathbf{x} \log p_t(\mathbf{x})$. We show that for the 1-step training and sufficiently small $\Delta t$, we minimize the score matching objective (6).

**Theorem 3.1.** *Consider a data set with trajectories sampled from SDE* (1) *and let* $\tilde{\mathcal{P}}^{-1}(\mathbf{x}) = -\mathcal{P}(\mathbf{x})$. *Then the 1-step loss* (3) *is equivalent to minimizing the score matching objective* (6) *as* $\Delta t \to 0$.

*Proof.*    See appendix A.1    □

**Maximum likelihood and multi-step training**  Extending the single step training to multiple steps does not directly minimize the score matching objective, but we can still interpret the learned correction in a probabilistic sense. For denoising score matching, it is possible to train $s_\theta$ via maximum likelihood training [Son+21a], which minimizes the KL-divergence between $p_0$ and the distribution obtained by sampling $\mathbf{x}_T$ from $p_T$ and simulating the probability flow ODE (8) from $t = T$ to $t = 0$. We derive a similar result for the multi-step loss.

**Theorem 3.2.** *Consider a data set with trajectories sampled from SDE* (1) *and let* $\tilde{\mathcal{P}}^{-1}(\mathbf{x}) = -\mathcal{P}(\mathbf{x})$. *Then the multi-step loss* (4) *maximizes a variational lower bound for maximum likelihood training of the probability flow ODE* (7) *as* $\Delta t \to 0$.

*Proof.*    See appendix A.2    □

To conclude, we have formulated a probabilistic multi-step training to solve inverse physics problems and provided a theoretical basis to solve these problems with score matching. Next, we outline additional details for implementing SMDP.

### 3.3   Training and Inference

We start training $s_\theta$ with the multi-step loss and window size $S = 2$, which is equivalent to the 1-step loss. Then, we gradually increase the window size $S$ until a maximum $S_{\max}$. For $S > 2$, the unrolling of the predicted trajectory includes interactions between $s_\theta$ and the reverse physics simulator $\tilde{\mathcal{P}}^{-1}$. For inference, we consider the neural SDE

$$d\mathbf{x} = \left[ -\tilde{\mathcal{P}}^{-1}(\mathbf{x}) - C\, s_\theta(\mathbf{x}, t) \right] dt + g(t)dW, \tag{9}$$

which we solve via the Euler-Maruyama method. For $C = 2$, we obtain the system's reverse-time SDE and sampling from this SDE yields the posterior distribution. Setting $C = 1$ and excluding the noise gives the probability flow ODE (8). We denote the ODE variant by *SMDP ODE* and the SDE variant by *SMDP SDE*. While the SDE can be used to obtain many samples and to explore the posterior distribution, the ODE variant constitutes a unique and deterministic solution based on maximum likelihood training.

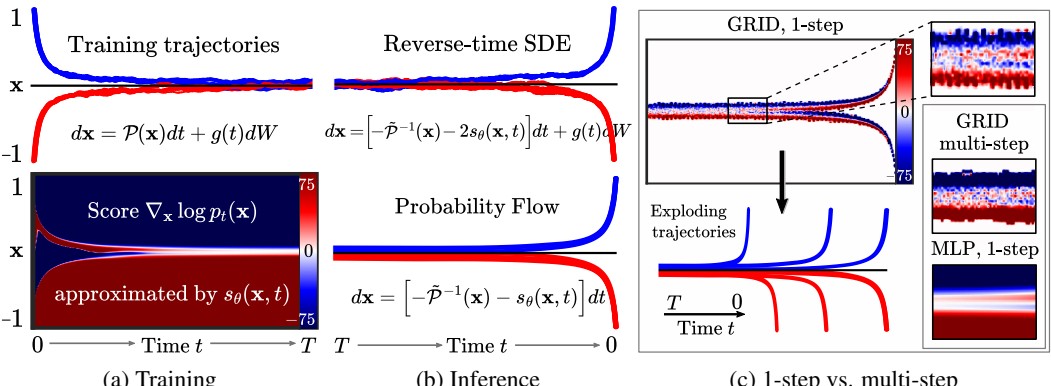

Figure 2: Overview of our 1D toy SDE. (a) Training with a data set of trajectories and known temporal dynamics given by $\mathcal{P}(\mathbf{x}) := -\text{sign}(\mathbf{x})\mathbf{x}^2$ and $g \equiv 0.1$. We estimate the score $\nabla_\mathbf{x} \log p_t(\mathbf{x})$ with our proposed method using an MLP network for $s_\theta(\mathbf{x}, t)$. Negative values (blue) push down the trajectories, and positive ones (red) push them up. Together with the dynamics, this can be used to reverse the system as shown in (b) either with the reverse-time SDE or the probability flow ODE. A successful inversion of the dynamics requires the network $s_\theta$ to be robust and extrapolate well (c). Inference using GRID trained with the 1-step loss causes trajectories to explode, as the network does not extrapolate well. Training GRID with the multi-step loss solves this issue.

## 4 Experiments

We show the capabilities of the proposed algorithm with a range of experiments. The first experiment in section 4.1 uses a simple 1D process to compare our method to existing score matching baselines. The underlying model has a known posterior distribution which allows for an accurate evaluation of the performance, and we use it to analyze the role of the multi-step loss formulation. Secondly, in section 4.2, we experiment with the stochastic heat equation. This is a particularly interesting test case as the diffusive nature of the equation effectively destroys information over time. In section 4.3, we apply our method to a scenario without stochastic perturbations in the form of a buoyancy-driven Navier-Stokes flow with obstacles. This case highlights the usefulness of the ODE variant. Finally, in section 4.4, we consider the situation where the reverse physics simulator $\tilde{\mathcal{P}}^{-1}$ is not known. Here, we train a surrogate model $\tilde{\mathcal{P}}^{-1}$ for isotropic turbulence flows and evaluate how well SMDP works with a learned reverse physics simulator.

### 4.1 1D Toy SDE

As a first experiment with a known posterior distribution we consider a simple quadratic SDE of the form: $dx = -\left[\lambda_1 \cdot \text{sign}(x)x^2\right]dt + \lambda_2 dW$, with $\lambda_1 = 7$ and $\lambda_2 = 0.03$. Throughout this experiment, $p_0$ is a categorical distribution, where we draw either 1 or $-1$ with the same probability. The reverse-time SDE that transforms the distribution $p_T$ of values at $T = 10$ to $p_0$ is given by

$$dx = -\left[\lambda_1 \cdot \text{sign}(x)x^2 - \lambda_2^2 \cdot \nabla_x \log p_t(x)\right]dt + \lambda_2 dw. \tag{10}$$

In figure 2a, we show paths from this SDE simulated with the Euler-Maruyama method. The trajectories approach 0 as $t$ increases. Given the trajectory value at $t = 10$, it is no longer possible to infer the origin of the trajectory at $t = 0$.

This experiment allows us to use an analytic reverse simulator: $\tilde{\mathcal{P}}^{-1}(x) = \lambda_1 \cdot \text{sign}(x)x^2$. This is a challenging problem because the reverse physics step increases quadratically with $x$, and $s_\theta$ has to control the reverse process accurately to stay within the training domain, or paths will explode to infinity. We evaluate each model based on how well the predicted trajectories $\hat{x}_{0:T}$ match the posterior distribution. When drawing $\hat{x}_T$ randomly from $[-0.1, 0.1]$, we should obtain trajectories with $\hat{x}_0$ being either $-1$ or 1 with the same likelihood. We assign the label $-1$ or 1 if the relative distance of an endpoint is $< 10\%$ and denote the percentage in each class by $\rho_{-1}$ and $\rho_1$. As some trajectories miss the target, typically $\rho_{-1} + \rho_1 < 1$. Hence, we define the posterior metric $Q$ as twice the minimum of $\rho_{-1}$ and $\rho_1$, i.e., $Q := 2 \cdot \min(\rho_{-1}, \rho_1)$ so that values closer to one indicate a better match with the correct posterior distribution.

**Training**  The training data set consists of 2500 simulated trajectories from 0 to $T$ and $\Delta t = 0.02$. Therefore each training trajectory has a length of $M = 500$. For the network $s_\theta(x, t)$, we consider a multilayer perceptron (MLP) and, as a special case, a grid-based discretization (GRID). The latter is not feasible for realistic use cases and high-dimensional data but provides means for an in-depth analysis of different training variants. For GRID, we discretize the domain $[0, T] \times [-1.25, 1.25]$ to obtain a rectangular grid with $500 \times 250$ cells and linearly interpolate the solution. The cell centers are initialized with 0. We evaluate $s_\theta$ trained via the 1-step and multi-step losses with $S_{\max} = 10$. Details of hyperparameters and model architectures are given in appendix C.

**Better extrapolation and robustness from multi-step loss**  See figure 2c for an overview of the differences between the learned score from MLP and GRID and the effects of the multi-step loss. For the 1-step training with MLP, we observe a clear and smooth score field with two tubes that merge to one at $x = 0$ as $t$ increases. As a result, the trajectories of the probability flow ODE and reverse-time SDE converge to the correct value. Training via GRID shows that most cells do not get any gradient updates and remain 0. This is caused by a need for more training data in these regions. In addition, the boundary of the trained region is jagged and diffuse. Trajectories traversing these regions can quickly explode. In contrast, the multi-step loss leads to a consistent signal around the center line at $x = 0$, effectively preventing exploding trajectories.

**Evaluation and comparison with baselines**  As a baseline for learning the scores, we consider implicit score matching [Hyv05, ISM]. Additionally, we consider sliced score matching with variance reduction [Son+19, SSM-VR] as a variant of ISM. We train all methods with the same network architecture using three different data set sizes. As can be seen in table 1, the 1-step loss, which is conceptually similar to denoising score matching, compares favorably against ISM and SSM-VR. All methods perform well for the reverse-time SDE, even for very little training data. Using the multi-step loss consistently gives significant improvements at the cost of a slightly increased training time. Our proposed multi-step training performs best or is on par with the baselines for all data set sizes and inference types. Because the posterior metric Q is very sensitive to the score where the paths from both starting points intersect, evaluations are slightly noisy.

| Method | Probability flow ODE | | | Reverse-time SDE | | |
|---|---|---|---|---|---|---|
| | Data set size | | | Data set size | | |
| | 100% | 10% | 1% | 100% | 10% | 1% |
| multi-step | **0.97** | **0.91** | **0.81** | **0.99** | 0.94 | **0.85** |
| 1-step | 0.78 | 0.44 | 0.41 | 0.93 | 0.71 | 0.75 |
| ISM | 0.19 | 0.15 | 0.01 | 0.92 | 0.94 | 0.52 |
| SSM-VR | 0.17 | 0.49 | 0.27 | 0.88 | 0.94 | 0.67 |

Table 1: Posterior metric $Q$ for 1000 predicted trajectories averaged over three runs. For standard deviations, see table 3 in the appendix.

**Comparison with analytic scores**  We perform further experiments to empirically verify Theorem 3.1 by comparing the learned scores of our method with analytic scores in appendix C.

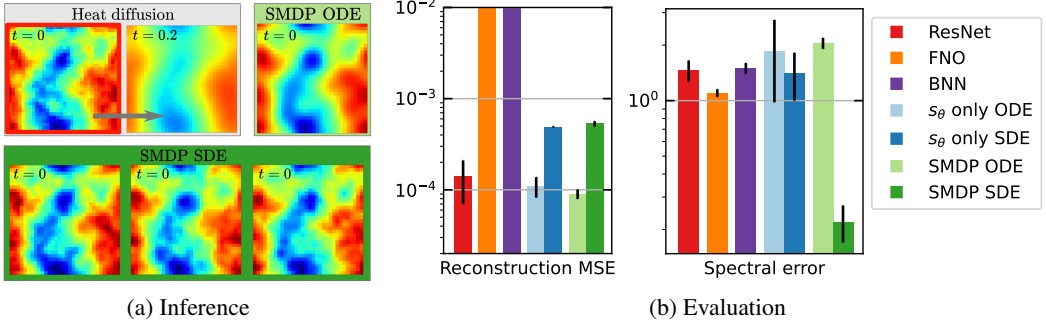

(a) Inference  (b) Evaluation

Figure 3: Stochastic heat equation overview. While the ODE trajectories provide smooth solutions with the lowest reconstruction MSE, the SDE solutions synthesize high-frequency content, significantly improving spectral error. The "$s_\theta$ only" version without the reverse physics step exhibits a significantly larger spectral error. Metrics in (b) are averaged over three runs.

## 4.2 Stochastic Heat Equation

The heat equation $\frac{\partial u}{\partial t} = \alpha \Delta u$ plays a fundamental role in many physical systems. For this experiment, we consider the stochastic heat equation, which slightly perturbs the heat diffusion process and includes an additional term $g(t)\,\xi$, where $\xi$ is space-time white noise, see Pardoux [Par21, Chapter 3.2]. For our experiments, we fix the diffusivity constant to $\alpha = 1$ and sample initial conditions at $t = 0$ from Gaussian random fields with $n = 4$ at resolution $32 \times 32$. We simulate the heat diffusion with noise from $t = 0$ until $t = 0.2$ using the Euler-Maruyama method and a spectral solver $\mathcal{P}_h$ with a fixed step size $\Delta t = 6.25 \times 10^{-3}$ and $g \equiv 0.1$. Given a simulation end state $\mathbf{x}_T$, we want to recover a possible initial state $\mathbf{x}_0$. In this experiment, the forward solver cannot be used to infer $\mathbf{x}_0$ directly in a single step or without corrections since high frequencies due to noise are amplified, leading to physically implausible solutions. We implement the reverse physics simulator $\tilde{P}^{-1}$ by using the forward step of the solver $\mathcal{P}_h(\mathbf{x})$, i.e. $\tilde{\mathcal{P}}^{-1}(\mathbf{x}) \approx -\mathcal{P}_h(\mathbf{x})$.

**Training and Baselines**   Our training data set consists of 2500 initial conditions with their corresponding trajectories and end states at $t = 0.2$. We consider a small *ResNet*-like architecture based on an encoder and decoder part as representation for the score function $s_\theta(\mathbf{x}, t)$. The spectral solver is implemented via differentiable programming in *JAX* [SC20], see appendix D. As baseline methods, we consider a supervised training of the same *ResNet*-like architecture as $s_\theta(\mathbf{x}, t)$, a *Bayesian neural network* (BNN) as well as a *Fourier neural operator* (FNO) network [Li+21]. We adopt an $L_2$ loss for all these methods, i.e., the training data consists of pairs of initial state $\mathbf{x}_0$ and end state $\mathbf{x}_T$.

Additionally, we consider a variant of our proposed method for which we remove the reverse physics step $\tilde{\mathcal{P}}^{-1}$ such that the inversion of the dynamics has to be learned entirely by $s_\theta$, denoted by "$s_\theta$ only". We do not compare to ISM and SSM-VR in the following as the data dimensions are too high for both methods to train properly, and we did not obtain stable trajectories during inference.

**Reconstruction accuracy vs. fitting the data manifold**   We evaluate our method and the baselines by considering the *reconstruction MSE* on a test set of 500 initial conditions and end states. For the reconstruction MSE, we simulate the prediction of the network forward in time with the solver $\mathcal{P}_h$ to obtain a corresponding end state, which we compare to the ground truth via the $L_2$ distance. This metric has the disadvantage that it does not measure how well the prediction matches the training data manifold. I.e., for this case, whether the prediction resembles the properties of the Gaussian random field. For that reason, we additionally compare

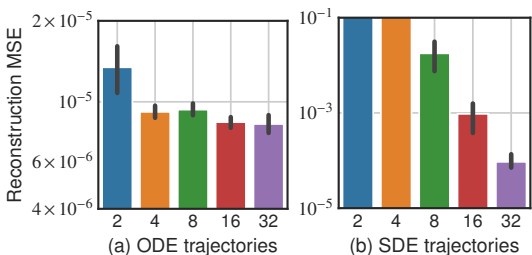

Figure 4: Multi-step $S_{\max}$ vs. reconstruction MSE averaged over 5 runs.

the power spectral density of the states as the *spectral loss*. An evaluation and visualization of the reconstructions are given in figure 3, which shows that our ODE inference performs best regarding the reconstruction MSE. However, its solutions are smooth and do not contain the necessary small-scale structures. This is reflected in a high spectral error. The SDE variant, on the other hand, performs very well in terms of spectral error and yields visually convincing solutions with only a slight increase in the reconstruction MSE. This highlights the role of noise as a source of entropy in the inference process for SMDP SDE, which is essential for synthesizing small-scale structures. Note that there is a natural tradeoff between both metrics, and the ODE and SDE inference perform best for each of the cases while using an identical set of weights.

**Multi-step loss is crucial for good performance**   We performed an ablation study on the maximum window size $S_{\max}$ in figure 4 for the reconstruction MSE. For both ODE and SDE inference, increasing $S_{\max}$ yields significant improvements at the cost of slightly increased training resources. This also highlights the importance of using a multi-step loss instead of the 1-step loss ($S_{\max} = 2$) for inverse problems with poor conditioning.

We perform further experiments regarding test-time distribution shifts when modifying the noise scale and diffusivity, see appendix D, which showcase the robustness of our methods.

### 4.3 Buoyancy-driven Flow with Obstacles

Next, we test our methodology on a more challenging problem. For this purpose, we consider deterministic simulations of buoyancy-driven flow within a fixed domain $\Omega \subset [0, 1] \times [0, 1]$ and randomly placed obstacles. Each simulation runs from time $t = 0.0$ to $t = 0.65$ with a step size of $\Delta t = 0.01$. SMDP is trained with the objective of reconstructing a plausible initial state given an end state of the marker density and velocity fields at time $t = 0.65$, as shown in figure 5a and figure 5b. We place spheres and boxes with varying sizes at different positions within the simulation domain that do not overlap with the marker inflow. For each simulation, we place one to two objects of each category.

**Score matching for deterministic systems** During training, we add Gaussian noise to each simulation state $\mathbf{x}_t$ with $\sigma_t = \sqrt{\Delta t}$. In this experiment, no stochastic forcing is used to create the data set, i.e., $g \equiv 0$. By adding noise to the simulation states, the 1-step loss still minimizes a score matching objective in this situation, similar to denoising score matching; see appendix A.3 for a derivation. In the situation without stochastic forcing, during inference, our method effectively alternates between the reverse physics step, a small perturbation, and the correction by $s_\theta(\mathbf{x}, t)$, which projects the perturbed simulation state back to the distribution $p_t$. We find that for the SDE trajectories, $C = 2$ slightly overshoots, and $C = 1$ gives an improved performance. In this setting, the "$s_\theta$ only" version of our method closely resembles a denoiser that learns additional physics dynamics.

**Training and comparison** Our training data set consists of 250 simulations with corresponding trajectories generated with *phiflow* [Hol+20]. Our neural network architecture for $s_\theta(\mathbf{x}, t)$ uses dilated convolutions [Sta+21], see appendix E for details. The reverse physics step $\tilde{\mathcal{P}}^{-1}$ is implemented directly in the solver by using a negative step size $-\Delta t$ for time integration. For training, we consider the multi-step formulation with $S_{\max} = 20$. We additionally compare with solutions from directly optimizing the initial smoke and velocity states at $t = 0.35$ using the differentiable forward simulation and limited-memory BFGS [LN89, LBFGS]. Moreover, we compare with solutions obtained from diffusion posterior sampling for general noisy inverse problems [Chu+23, DPS] with a pretrained diffusion model on simulation states at $t = 0.35$. For the evaluation, we consider a reconstruction MSE analogous to section 4.2 and the perceptual similarity metric LPIPS. The test set contains five simulations. The SDE version yields good results for this experiment but is most likely constrained in performance by the approximate reverse physics step and large step sizes. However, the ODE version outperforms directly inverting the simulation numerically ($\tilde{\mathcal{P}}^{-1}$ only), and when training without the reverse physics step ($s_\theta$ only), as shown in 5c.

### 4.4 Navier-Stokes with Unknown Dynamics

As a fourth experiment, we aim to learn the time evolution of isotropic, forced turbulence with a similar setup as Li et al. [Li+21]. The training data set consists of vorticity fields from 1000 simulation trajectories from $t = 0$ until $T = 10$ with $\Delta t = 1$, a spatial resolution of $64 \times 64$ and

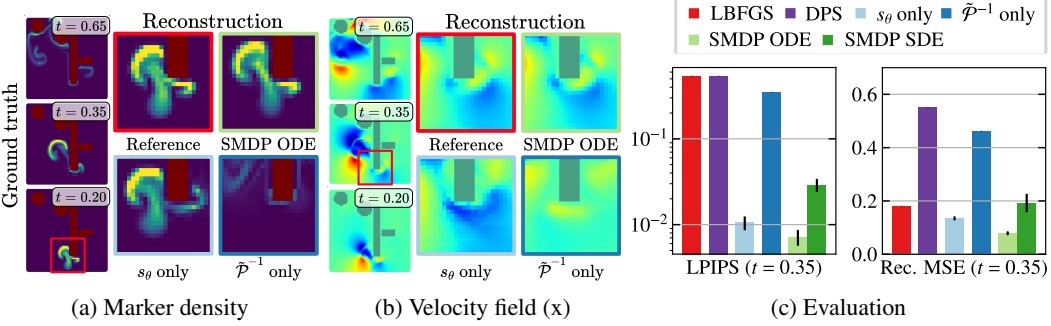

Figure 5: Buoyancy flow case. Ground truth shows the marker density and velocity field in the $x$-direction at different points of the simulation trajectory from the test set (a, b). We show reconstructions given the simulation end state at $t = 0.65$ and provide an evaluation of the reconstructed trajectories based on perceptual similarity (LPIPS) and the reconstruction MSE for three runs (c).

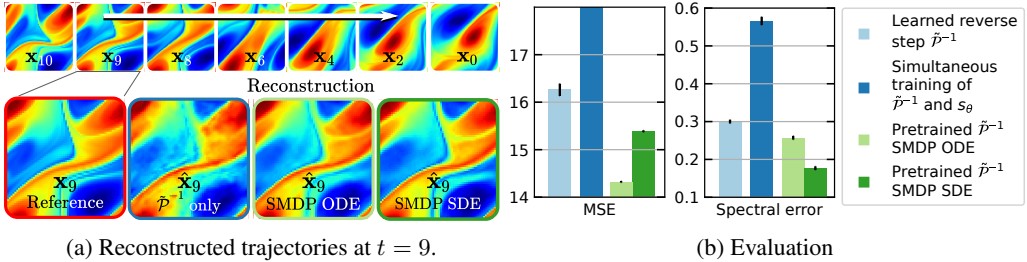

(a) Reconstructed trajectories at $t = 9$.    (b) Evaluation

Figure 6: Turbulence case. Comparison of reconstructed trajectories (a) and evaluation of MSE and spectral error for different training variants (b). Our proposed ODE and SDE inference outperforms the learned surrogate model $\tilde{\mathcal{P}}^{-1}$. Metrics are averaged over three runs.

viscosity fixed at $\nu = 10^{-5}$. As before, our objective is to predict a trajectory $\hat{\mathbf{x}}_{0:M}$ that reconstructs the true trajectory given an end state $\mathbf{x}_M$. In this experiment, we pretrain a surrogate for the reverse physics step $\tilde{\mathcal{P}}^{-1}$ by employing the FNO architecture from [Li+21] trained on the reverse simulation. For pretraining $\tilde{\mathcal{P}}^{-1}$ we use our proposed training setup with the multi-step loss and $S_{\max} = 10$ but freeze the score to $s_\theta(\mathbf{x}, t) \equiv 0$. Then, we train the time-dependent score $s_\theta(\mathbf{x}, t)$ while freezing the reverse physics step. This approach guarantees that any time-independent physics are captured by $\tilde{\mathcal{P}}^{-1}$ and $s_\theta(\mathbf{x}, t)$ can focus on learning small improvements to $\tilde{\mathcal{P}}^{-1}$ as well as respond to possibly time-dependent data biases. We give additional training details in appendix F.

**Evaluation and training variants**   For evaluation, we consider the MSE and spectral error of the reconstructed initial state $\hat{\mathbf{x}}_0$ compared to the reference $\mathbf{x}_0$. As baselines, during inference, we employ only the learned surrogate model $\tilde{\mathcal{P}}^{-1}$ without $s_\theta$. In addition to that, we evaluate a variant for which we train both the surrogate model and $s_\theta(\mathbf{x}, t)$ at the same time. As the two components resemble the drift and score of the reverse-time SDE, this approach is similar to *DiffFlow* [ZC21], which learns both components in the context of generative modeling. We label this approach *simultaneous training*. Results are shown in figure 6. Similar to the stochastic heat equation results in section 4.2, the SDE achieves the best spectral error, while the ODE obtains the best MSE. Our proposed method outperforms both the surrogate model and the simultaneous training of the two components.

## 5   Discussion and Conclusions

We presented a combination of learned corrections training and diffusion models in the context of physical simulations and differentiable physics for solving inverse physics problems. We showed its competitiveness, accuracy, and long-term stability in challenging and versatile experiments and motivated our design choices. We considered two variants with complementary benefits for inference: while the ODE variants achieve the best MSE, the SDE variants allow for sampling the posterior and yield an improved coverage of the target data manifold. Additionally, we provided theoretical insights that the 1-step is mathematically equivalent to optimizing the score matching objective. We showed that its multi-step extension maximizes a variational lower bound for maximum likelihood training.

Despite the promising initial results, our work has limitations that open up exciting directions for future work: Among others, it requires simulating the system backward in time step by step, which can be costly and alleviated by reduced order methods. Additionally, we assume that $\Delta t$ is sufficiently small and the reverse physics simulator is accurate enough. Determining a good balance between accurate solutions with few time steps and diverse solutions with many time steps represents an important area for future research.

**Acknowledgements**   BH, SV, and NT acknowledge funding from the European Research Council (ERC) under the European Union's Horizon 2020 research and innovation programme (LEDA: grant agreement No 758853, and SpaTe: grant agreement No 863850). SV thanks the Max Planck Society for support through a Max Planck Lise Meitner Group. This research was partly carried out on the High Performance Computing resources of the FREYA cluster at the Max Planck Computing and Data Facility (MPCDF) in Garching operated by the Max Planck Society (MPG).

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

# Appendix

## A  Proofs and Training Methodology

Below we summarize the problem formulation from the main paper and provide details about the training procedure and the derivation of our methodology.

**Problem setting**  Let $(\Omega, \mathcal{F}, P)$ be a probability space and $W(t) = (W_1(t), ..., W_D(t))^T$ be a $D$-dimensional Brownian motion. Moreover, let $\mathbf{x}_0$ be a $\mathcal{F}_0$-measurable $\mathbb{R}^D$-valued random variable that is distributed as $p_0$ and represents the initial simulation state. We consider the time evolution of the physical system for $0 \leq t \leq T$ modeled by the stochastic differential equation (SDE)

$$d\mathbf{x} = \mathcal{P}(\mathbf{x})dt + g(t)dW \tag{11}$$

with initial value $\mathbf{x}_0$ and Borel measurable drift $\mathcal{P} : \mathbb{R}^D \to \mathbb{R}^D$ and diffusion $g : [0, T] \to \mathbb{R}_{\geq 0}$. This SDE transforms the marginal distribution $p_0$ of initial states at time 0 to the marginal distribution $p_T$ of end states at time $T$.

Moreover, we assume that we have sampled $N$ trajectories of length $M$ from the above SDE with a fixed time discretization $0 \leq t_0 < t_1 < ... < t_M \leq T$ for the interval $[0, T]$ and collected them in a training data set $\{(\mathbf{x}_{t_m}^{(n)})_{i=0}^M\}_{n=0}^N$. For simplicity, we assume that all time steps are equally spaced, i.e., $t_{m+1} - t_m := \Delta t$. Moreover, in the following we use the notation $\mathbf{x}_{\ell:m}$ for $0 \leq \ell < m \leq M$ to refer to the trajectory $(\mathbf{x}_{t_\ell}, \mathbf{x}_{t_{\ell+1}}, ..., \mathbf{x}_{t_m})$.

**Assumptions**  Throughout this paper, we make some additional assumptions to ensure the existence of a unique solution to the SDE (11) and the strong convergence of the Euler-Maruyama method. In particular:

- Finite variance of samples from $p_0$: $\mathbb{E}_{\mathbf{x}_0 \sim p_0}[||\mathbf{x}_0||_2^2] < \infty$
- Lipschitz continuity of $\mathcal{P}$: $\exists K_1 > 0 \, \forall \mathbf{x}, \mathbf{y} \in \mathbb{R}^D : ||\mathcal{P}(\mathbf{x}) - \mathcal{P}(\mathbf{y})||_2 \leq K_1 ||\mathbf{x} - \mathbf{y}||_2$
- Lipschitz continuity of $g$: $\exists K_2 > 0 \, \forall t, s \in [0, T] : |g(t) - g(s)| \leq K_3 |t - s|$
- Linear growth condition: $\exists K_3 > 0 \, \forall \mathbf{x} \in \mathbb{R}^D : ||\mathcal{P}(\mathbf{x})||_2 \leq K_3(1 + ||\mathbf{x}||_2)$
- $g$ is bounded: $\exists K_4 > 0 \, \forall t \in [0, T] : |g(t)| \leq K_4$

**Euler-Maruyama Method**  Using Euler-Maruyama steps, we can simulate paths from SDE (11) similar to ordinary differential equations (ODE). Given an initial state $\mathbf{X}_{t_0}$, we let $\hat{\mathbf{X}}_{t_0}^{\Delta t} = \mathbf{X}_{t_0}$ and define recursively

$$\hat{\mathbf{X}}_{t_{m+1}}^{\Delta t} \leftarrow \hat{\mathbf{X}}_{t_m}^{\Delta t} + \Delta t \, \mathcal{P}(\hat{\mathbf{X}}_{t_m}^{\Delta t}) + \sqrt{\Delta t} \, g(t_m) \, z_{t_m}, \tag{12}$$

where $z_{t_m}$ are i.i.d. with $z_{t_m} \sim \mathcal{N}(0, I)$. For $t_i \leq t < t_{i+1}$, we define the piecewise constant solution of the Euler-Maruyama Method as $\hat{\mathbf{X}}_t^{\Delta t} := \hat{\mathbf{X}}_{t_i}^{\Delta t}$. Let $\mathbf{X}_t$ denote the solution of the SDE (11). Then the Euler-Maruyama solution $\hat{\mathbf{X}}_t^{\Delta t}$ converges strongly to $\mathbf{X}_t$.

**Lemma A.1.** *[Strong convergence of Euler-Maruyama method] Consider the piecewise constant solution $\hat{X}_t^{\Delta t}$ of the Euler-Maruyama method. There is a constant $C$ such that*

$$\sup_{0 \leq t \leq T} \mathbb{E}[||X_t - \hat{X}_t^{\Delta t}||_2] \leq C\sqrt{\Delta t}. \tag{13}$$

*Proof.*  See Kloeden et al. [Klo+92, p. 10.2.2]  □

### A.1  1-step Loss and Score Matching Objective

**Theorem A.2.** *Consider a data set $\{\mathbf{x}_{0:m}^{(n)}\}_{n=1}^N$ with trajectories sampled from SDE (11). Then the 1-step loss*

$$\mathcal{L}_{\text{single}}(\theta) := \frac{1}{M} \mathbb{E}_{\mathbf{x}_{0:M}} \left[ \sum_{m=0}^{M-1} \left[ ||\mathbf{x}_m - \{\mathbf{x}_{m+1} + \Delta t \left[ -\mathcal{P}(\mathbf{x}_{m+1}) + s_\theta(\mathbf{x}_{m+1}, t_{m+1}) \right] \}||^2 \right] \right] \tag{14}$$

*is equivalent to minimizing the score matching objective*

$$\mathcal{J}(\theta) := \int_0^T \mathbb{E}_{\mathbf{x}_t}[||\nabla_{\mathbf{x}} \log p_t(\mathbf{x}) - \tilde{s}_\theta(\mathbf{x}, t)||_2^2] dt, \tag{15}$$

*where $\tilde{s}_\theta(\mathbf{x}, t) = s_\theta(\mathbf{x}, t)/g^2(t)$ as $\Delta t \to 0$.*

*Proof.*

"$\Leftarrow$": Consider $\theta^*$ such that $\tilde{s}_{\theta^*}(\mathbf{x}, t) \equiv \nabla_{\mathbf{x}} \log p_t(\mathbf{x})$, which minimizes the score matching objective $\mathcal{J}(\theta)$. Then fix a time step $t$ and sample $\mathbf{x}_t$ and $\mathbf{x}_{t+\Delta t}$ from the data set. The probability flow solution $\mathbf{x}_t^p$ based on equation (14) is

$$\mathbf{x}_t^p := \mathbf{x}_{t+\Delta t} + \Delta t \left[ -\mathcal{P}(\mathbf{x}_{t+\Delta t}) + s_\theta(\mathbf{x}_{t+\Delta t}, t + \Delta t) \right]. \tag{16}$$

At the same time, we know that the transformation of marginal likelihoods from $p_{t+\Delta t}$ to $p_t$ follows the reverse-time SDE [And82]

$$d\mathbf{x} = \left[ \mathcal{P}(\mathbf{x}) + g^2(t)\nabla_{\mathbf{x}} \log p_t(\mathbf{x}) \right] dt + g(t)dW, \tag{17}$$

which runs backward in time from $T$ to $0$. Denote by $\hat{\mathbf{x}}_t^{\Delta t}$ the solution of the Euler-Maruyama method at time $t$ initialized with $\mathbf{x}_{t+\Delta t}$ at time $t + \Delta t$.

Using the triangle inequality for squared norms, we can write

$$\lim_{\Delta t \to 0} \mathbb{E}\left[ ||\mathbf{x}_t - \mathbf{x}_t^p||_2^2 \right] \le 2 \lim_{\Delta t \to 0} \mathbb{E}\left[ ||\mathbf{x}_t - \hat{\mathbf{x}}_t^{\Delta t}||_2^2 \right] + 2 \lim_{\Delta t \to 0} \mathbb{E}\left[ ||\hat{\mathbf{x}}_t^{\Delta t} - \mathbf{x}_t^p||_2^2 \right]. \tag{18}$$

Because of the strong convergence of the Euler-Maruyama method, we have that for the first term of the bound in equation (18)

$$\lim_{\Delta t \to 0} \mathbb{E}\left[ ||\mathbf{x}_t - \hat{\mathbf{x}}_t^{\Delta t}||_2^2 \right] = 0 \tag{19}$$

independent of $\theta$. At the same time, for the Euler-Maruyama discretization, we can write

$$\hat{\mathbf{x}}_t^{\Delta t} = \mathbf{x}_{t+\Delta t} + \Delta t \left[ -\mathcal{P}(\mathbf{x}_{t+\Delta t}) + g^2(t + \Delta t)\nabla_{\mathbf{x}} \log p_{t+\Delta t}(\mathbf{x}_{t+\Delta t}) \right] \tag{20}$$

$$+ g(t + \Delta t)\sqrt{\Delta t} z_{t+\Delta t}, \tag{21}$$

where $z_{t+\Delta t}$ is a standard Gaussian distribution, i.e., $z_{t+\Delta t} \sim \mathcal{N}(0, I)$. Therefore, we can simplify the second term of the bound in equation (18)

$$\lim_{\Delta t \to 0} \mathbb{E}\left[ ||\hat{\mathbf{x}}_t^{\Delta t} - \mathbf{x}_t^p||_2^2 \right] \tag{22}$$

$$= \lim_{\Delta t \to 0} \mathbb{E}_{\mathbf{x}_{t+\Delta t} \sim p_{t+\Delta t}, z \sim \mathcal{N}(0,I)} \left[ \left|\left| \Delta t \, g(t + \Delta t)^2 \left[ \nabla_{\mathbf{x}} \log p_{t+\Delta t}(\mathbf{x}_{t+\Delta t}) \right.\right.\right.\right. \tag{23}$$

$$\left.\left.\left.\left. - \tilde{s}_\theta(\mathbf{x}_{t+\Delta t}, t + \Delta t) \right] + g(t + \Delta t)\sqrt{\Delta t} \, z \right|\right|_2^2 \right]. \tag{24}$$

If $\theta^*$ minimizes the score matching objective, then $\tilde{s}_{\theta^*}(\mathbf{x}, t) \equiv \nabla_{\mathbf{x}} \log p_t(\mathbf{x})$, and therefore the above is the same as

$$\lim_{\Delta t \to 0} \mathbb{E}_z[||\sqrt{\Delta t} \, g(t + \Delta t) \, z||_2^2] = 0. \tag{25}$$

Combining equations (18), (19) and (25) yields

$$\lim_{\Delta t \to 0} \mathbb{E}\left[ ||\mathbf{x}_t - \mathbf{x}_t^p||_2^2 \right] = 0. \tag{26}$$

Additionally, since $g$ is bounded, we even have

$$\mathbb{E}[||\sqrt{\Delta t} \, g(t + \Delta t) \, z||_2^2] \le \mathbb{E}[||\sqrt{\Delta t} \, K_4 \, z||_2^2] = \mathbb{E}[||K_4 \, z||_2^2]\Delta t. \tag{27}$$

For $\mathcal{L}_{\text{single}}(\theta)$, using the above bound (27) and strong convergence of the Euler-Maruyama method, we can therefore derive the following bound

$$\mathcal{L}_{\text{single}}(\theta) = \frac{1}{M} \mathbb{E}\left[ \sum_{m=0}^{M-1} \left[ ||\mathbf{x}_m + \Delta t \left[ -\mathcal{P}(\mathbf{x}_{m+1}) + s_\theta(\mathbf{x}_{m+1}, t_{m+1}) \right]||^2 \right] \right] \tag{28}$$

$$\le \frac{1}{M} M \, 2 \left[ C\sqrt{\Delta t} + \mathbb{E}_z[||K_4 \, z||_2^2]\Delta t \right], \tag{29}$$

which implies that $\mathcal{L}_{\text{single}}(\theta) \to 0$ as $\Delta t \to 0$.

"⇒": With the definitions from "⇐", let $\theta^*$ denote a minimizer such that $\mathcal{L}_{\text{single}}(\theta) \to 0$ as $\Delta t \to 0$, i.e., we assume there is a sequence $\Delta t_1, \Delta t_2, ...$ with $\lim_{n \to \infty} \Delta t_n = 0$ and a sequence $\theta_1, \theta_2, ...$, where $\theta_n$ is a global minimizer to the objective $\mathcal{L}_{\text{single}}^{\Delta t_n}(\theta)$ that depends on the step size $\Delta t_n$. If there is $\theta^*$ such that $s_{\theta^*}(\mathbf{x}, t) \equiv \nabla_{\mathbf{x}} \log p_t(\mathbf{x})$, then $\mathcal{L}_{\text{single}}^{\Delta t_n}(\theta_n) \leq \mathcal{L}_{\text{single}}^{\Delta t_n}(\theta^*)$. From "⇐" we know that $\lim_{n \to \infty} \mathcal{L}_{\text{single}}^{\Delta t_n}(\theta^*) = 0$ and therefore $\lim_{n \to \infty} \mathcal{L}_{\text{single}}^{\Delta t_n}(\theta_n) = 0$. This implies that each summand of $\mathcal{L}_{\text{single}}(\theta)$ also goes to zero as $\Delta t \to 0$, i.e., $\lim_{\Delta t \to 0} \mathbb{E}\left[||\mathbf{x}_t - \mathbf{x}_t^p||_2^2\right] = 0$. Again, with the triangle inequality for squared norms, we have that

$$\lim_{\Delta t \to 0} \mathbb{E}\left[||\hat{\mathbf{x}}_t^{\Delta t} - \mathbf{x}_t^p||_2^2\right] \leq 2 \lim_{\Delta t \to 0} \mathbb{E}\left[||\mathbf{x}_t - \hat{\mathbf{x}}_t^{\Delta t}||_2^2\right] + 2 \lim_{\Delta t \to 0} \mathbb{E}\left[||\mathbf{x}_t - \mathbf{x}_t^p||_2^2\right]. \quad (30)$$

By the strong convergence of the Euler-Maruyama method and $\theta = \theta^*$, we obtain

$$\lim_{\Delta t \to 0} \mathbb{E}\left[||\hat{\mathbf{x}}_t^{\Delta t} - \mathbf{x}_t^p||_2^2\right] = 0. \quad (31)$$

At the same time, for fixed $\Delta t > 0$, we can compute

$$\mathbb{E}\left[||\hat{\mathbf{x}}_t^{\Delta t} - \mathbf{x}_t^p||_2^2\right] \quad (32)$$

$$= \mathbb{E}_{\mathbf{x}_{t+\Delta t}, z \sim \mathcal{N}(0,I)}[||\Delta t \, g(t + \Delta t)^2 \left[\nabla_{\mathbf{x}} \log p_{t+\Delta t}(\mathbf{x}_{t+\Delta t})\right. \quad (33)$$

$$\left. - s_\theta(\mathbf{x}_{t+\Delta t}, t + \Delta t)\right] + \sqrt{\Delta t} \, g(t + \Delta t) \, z||_2^2] \quad (34)$$

$$= \Delta t \, g(t + \Delta t) \, \mathbb{E}_{\mathbf{x}_{t+\Delta t}, z \sim \mathcal{N}(0,I)}[||\Delta t^{3/2} \, g(t + \Delta t)^{5/2} \left[\nabla_{\mathbf{x}} \log p_{t+\Delta t}(\mathbf{x}_{t+\Delta t})\right. \quad (35)$$

$$\left. - s_\theta(\mathbf{x}_{t+\Delta t}, t + \Delta t)\right] + z||_2^2]. \quad (36)$$

For fixed $\mathbf{x}_{t+\Delta t}$, the distribution over $z \sim \mathcal{N}(0, I)$ in equation (36) correspond to a non-central chi-squared distribution [JKB95, Chapter 13.4], whose mean can be calculated as

$$\mathbb{E}_{z \sim \mathcal{N}(0,I)}\left[\left|\left|\Delta t^{3/2} \, g(t + \Delta t)^{5/2} \left[\nabla_{\mathbf{x}} \log p_{t+\Delta t}(\mathbf{x}_{t+\Delta t}) - s_\theta(\mathbf{x}_{t+\Delta t}, t + \Delta t)\right] + z\right|\right|_2^2\right] \quad (37)$$

$$= \left|\left|\Delta t^{3/2} \, g(t + \Delta t)^{5/2} \left[\nabla_{\mathbf{x}} \log p_{t+\Delta t}(\mathbf{x}_{t+\Delta t}) - s_\theta(\mathbf{x}_{t+\Delta t}, t + \Delta t)\right]\right|\right|_2^2 + D. \quad (38)$$

For each $\Delta t > 0$, the above is minimized if and only if $\nabla_{\mathbf{x}} \log p_{t+\Delta t}(\mathbf{x}_{t+\Delta t}) = s_\theta(\mathbf{x}_{t+\Delta t}, t + \Delta t)$.

□

## A.2 Multi-step Loss and Maximum Likelihood Training

We now extend the 1-step formulation from above to multiple steps and discuss its relation to maximum likelihood training. For this, we consider our proposed probability flow ODE defined by

$$d\mathbf{x} = \left[\mathcal{P}(\mathbf{x}) + s_\theta(\mathbf{x}, t)\right] dt \quad (39)$$

and for $t_i < t_j$ define $p_{t_i}^{t_j, \text{ODE}}$ as the distribution obtained by sampling $\mathbf{x} \sim p_{t_j}$ and integrating the probability flow with network $s_\theta(\mathbf{x}, t)$ equation (7) backward in time until $t_i$. We can choose two arbitrary time points $t_i$ and $t_j$ with $t_i < t_j$ because we do not require fixed start and end times of the simulation.

The maximum likelihood training objective of the probability flow ODE (7) can be written as maximizing

$$\mathbb{E}_{\mathbf{x}_{t_i} \sim p_{t_i}}[\log p_{t_i}^{p_{t_j}, \text{ODE}}(\mathbf{x}_{t_i})]. \quad (40)$$

Our proposed multi-step loss is based on the sliding window size $S$, which is the length of the sub-trajectory that we aim to reconstruct with the probability flow ODE (7). The multi-step loss is defined as

$$\mathcal{L}_{\text{multi}}(\theta) := \frac{1}{M} \mathbb{E}_{\mathbf{x}_{0:M}}\left[\sum_{m=0}^{M-S+1} \left[||\mathbf{x}_{m:m+S-1} - \hat{\mathbf{x}}_{m:m+S-1}||_2^2\right]\right], \quad (41)$$

where we compute the expectation by sampling $\mathbf{x}_{0:m}$ from the training data set and $\hat{\mathbf{x}}_{i:i+S-1}$ is the predicted sub-trajectory that is defined recursively by

$$\hat{\mathbf{x}}_{i+S} = \mathbf{x}_{i+S} \quad \text{and} \quad \hat{\mathbf{x}}_{i+S-1-j} = \hat{\mathbf{x}}_{i+S-j} + \Delta t \left[ -\mathcal{P}(\hat{\mathbf{x}}_{i+S-j}) + s_\theta(\hat{\mathbf{x}}_{i+S-j}, t_{i+S-j}) \right]. \quad (42)$$

In the following, we show that the multi-step loss (4) maximizes a variational lower bound for the maximum likelihood training objective (40).

**Theorem A.3.** *Consider a data set $\{\mathbf{x}_{0:m}^{(n)}\}_{n=1}^N$ with trajectories sampled from SDE* (11)*. Then the multi-step loss* (4) *maximizes a variational lower bound for the maximum likelihood training objective of the probability flow ODE* (40) *as $\Delta t \to 0$.*

Let $\mu_{t_i}^{\mathrm{ODE}}(\mathbf{x}_{t_j})$ denote the solution of the probability flow ODE (7) integrated backward from time $t_j$ to $t_i$ with initial value $\mathbf{x}_{t_j}$.

For the maximum likelihood objective, we can derive a variational lower bound

$$\mathbb{E}_{\mathbf{x}_{t_i}} \left[ \log p_{t_i}^{t_j, \mathrm{ODE}}(\mathbf{x}_{t_i}) \right] = \mathbb{E}_{\mathbf{x}_{t_i}} \left[ \log \left( \mathbb{E}_{\mathbf{x}_{t_j}} \left[ p_{t_i}^{t_j, \mathrm{ODE}}(\mathbf{x}_{t_i} | \mathbf{x}_{t_j}) \right] \right) \right] \quad (43)$$

$$= \mathbb{E}_{\mathbf{x}_{t_i}} \left[ \log \left( \mathbb{E}_{\mathbf{x}_{t_j} | \mathbf{x}_{t_i}} \left[ \frac{p_{t_i}(\mathbf{x}_{t_i})}{p_{t_j}(\mathbf{x}_{t_j} | \mathbf{x_{t_i}})} p_{t_i}^{t_j, \mathrm{ODE}}(\mathbf{x}_{t_i} | \mathbf{x}_{t_j}) \right] \right) \right] \quad (44)$$

$$\geq \mathbb{E}_{\mathbf{x}_{t_i}} \mathbb{E}_{\mathbf{x}_{t_j} | \mathbf{x}_{t_i}} \left[ \log \left( \frac{p_{t_j}(\mathbf{x}_{t_j})}{p_{t_j}(\mathbf{x}_{t_j} | \mathbf{x_{t_i}})} p_{t_i}^{t_j, \mathrm{ODE}}(\mathbf{x}_{t_i} | \mathbf{x}_{t_j}) \right) \right] \quad (45)$$

$$= \mathbb{E}_{\mathbf{x}_{t_i}} \mathbb{E}_{\mathbf{x}_{t_j} | \mathbf{x}_{t_i}} \left[ \log \left( \frac{p_{t_j}(\mathbf{x}_{t_j})}{p_{t_j}(\mathbf{x}_{t_j} | \mathbf{x_{t_i}})} \right) + \log \left( p_{t_i}^{t_j, \mathrm{ODE}}(\mathbf{x}_{t_i} | \mathbf{x}_{t_j}) \right) \right], \quad (46)$$

where the inequality is due to Jensen's inequality. Since only $p_{t_i}^{t_j, \mathrm{ODE}}(\mathbf{x}_{t_i} | \mathbf{x}_{t_j})$ depends on $\theta$, this is the same as maximizing

$$\mathbb{E}_{\mathbf{x}_{t_i}} \mathbb{E}_{\mathbf{x}_{t_j} | \mathbf{x}_{t_i}} \left[ \log \left( p_{t_i}^{t_j, \mathrm{ODE}}(\mathbf{x}_{t_i} | \mathbf{x}_{t_j}) \right) \right]. \quad (47)$$

The probability flow ODE is likelihood-free, which makes it challenging to optimize. Therefore, we relax the objective by perturbing the ODE distributions by convolving them with a Gaussian kernel $G_\epsilon$ with small $\epsilon > 0$, see, e.g., Kersting et al. [Ker+20, Gaussian ODE filtering]. This allows us to model the conditional distribution $p_{t_i}^{t_j, \mathrm{ODE}} | \mathbf{x}_{t_j}$ as a Gaussian distribution with mean $\mu_{t_i}^{t_j, \mathrm{ODE}}(\mathbf{x}_{t_j})$ and variance $\sigma^2 = \epsilon$. Then maximizing (47) reduces to matching the mean of the distribution, i.e., minimizing

$$\mathbb{E}_{\mathbf{x}_{t_i}} \mathbb{E}_{\mathbf{x}_{t_j} | \mathbf{x}_{t_i}} \left[ ||\mathbf{x}_{t_i} - \mu_{t_i}^{\mathrm{ODE}}(\mathbf{x}_{t_j})||_2^2 \right] \quad (48)$$

independent of $\epsilon > 0$. Since this is true for any time step $t_j > t_i$ and corresponding simulation state $\mathbf{x}_{t_j}$ given $\mathbf{x}_{t_i}$, we can pick the pairs $(\mathbf{x}_{t_i}, \mathbf{x}_{t_{i+1}})$, $(\mathbf{x}_{t_i}, \mathbf{x}_{t_{i+2}})$, $(\mathbf{x}_{t_i}, \mathbf{x}_{t_{i+3}})$ and so on. Then, we can optimize them jointly by considering the sum of the individual objectives up to a maximum sliding window size

$$\mathbb{E}_{\mathbf{x}_{i:j}} \left[ \sum_{k=i}^{j-1} ||\mathbf{x}_{t_k} - \mu_{t_k}^{\mathrm{ODE}}(\mathbf{x}_{t_j})||_2^2 \right]. \quad (49)$$

As $\Delta t \to 0$, we compute the terms $\mu_{t_k}^{\mathrm{ODE}}(\mathbf{x}_{t_j})$ on a single trajectory starting at $\mathbf{x}_{t_i}$ with sliding window $S$ covering the trajectory until $\mathbf{x}_{t_j}$ via the Euler method, i.e., we can define recursively

$$\hat{\mathbf{x}}_{i+S} = \mathbf{x}_{i+S} \quad \text{and} \quad \hat{\mathbf{x}}_{i+S-1-j} = \hat{\mathbf{x}}_{i+S-j} + \Delta t \left[ -\mathcal{P}(\mathbf{x}_{i+S-j}) + s_\theta(\mathbf{x}_{i+S-j}, t_{i+S-j}) \right]. \quad (50)$$

By varying the starting points of the sliding window $\mathbf{x}_{t_i}$, this yields our proposed multi-step loss $\mathcal{L}_{\mathrm{multi}}(\theta)$.

## A.3   Denoising Score Matching for Deterministic Simulations

So far, we have considered physical systems that can be modeled by an SDE, i.e., equation (11). While this problem setup is suitable for many scenarios, we would also like to apply a similar

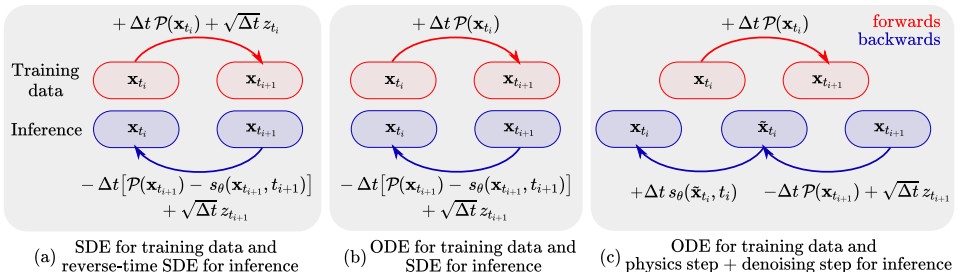

(a) SDE for training data and reverse-time SDE for inference

(b) ODE for training data and SDE for inference

(c) ODE for training data and physics step + denoising step for inference

Figure 7: Variants of training and inference for different physical systems. (a) shows the SDE and reverse-time SDE setup with the Euler-Maruyama discretization when the system is modeled by an SDE. The diffusion term $g(t)$ is absorbed in the Gaussian random variable $z_t \sim \mathcal{N}(0, g(t)^2 I)$ and network $s_\theta(\mathbf{x}, t)$. In (b), we assume that the temporal evolution of the training data is deterministic, i.e., we model the physical system without the diffusion term. However, for inference, we consider the reverse-time SDE of the same form as in (a), where the diffusion coefficient $g(t)$ is chosen as a hyperparameter that depends on the noise scale added to the data. Then, in (c), we split the Euler step for the backward direction into a physics-only update, adding the Gaussian noise $z$ and a denoising step by $s_\theta(\mathbf{x}, t)$.

methodology when the system is deterministic, i.e., when we can write the problem as an ordinary stochastic equation

$$d\mathbf{x} = \mathcal{P}(\mathbf{x})dt. \tag{51}$$

In the case of chaotic dynamical systems, this still represents a hard inverse problem, especially when information is lost due to noise added to the trajectories after their generation.

The training setup based on modeling the physics system with an SDE is shown in figure 7a. Figure 7b and 7c illustrate two additional data setup and inference variants for deterministic physical systems modeled by the ODE (51). While for the experiments in sections 3.1 and 3.2 in the main paper, our setup resembles (a), for the buoyancy-driven flow in section 3.3 and the forced isotropic turbulence in section 3.4 in the main paper, we consider (c) as the system is deterministic.

For this variant, the update by $-\mathcal{P}(\mathbf{x})$ and $s_\theta(\mathbf{x}, t)$ is separated into two steps. The temporal evolution from $t_{i+1}$ to $t_i$ is then defined entirely by physics. We apply an additive noise to the system and the update step by $s_\theta(\mathbf{x}, t)$, which can be interpreted as denoising for a now slightly perturbed state $\tilde{\mathbf{x}}_{t_i}$. In this case, we show that the network $s_\theta(\mathbf{x}, t)$ still learns the correct score $\nabla_\mathbf{x} \log p_t(\mathbf{x})$ during training using denoising score matching. We compare the performance of variants (b) and (c) for the buoyancy-drive flow in appendix E.

When separating physics and score updates, we calculate the updates as

$$\hat{\mathbf{x}}_{t_i} = \mathbf{x}_{t_{i+1}} - \Delta t \, \mathcal{P}(\mathbf{x}_{t_{i+1}}) \tag{52}$$

$$\hat{\mathbf{x}}_{t_i}^{\text{noise}} = \hat{\mathbf{x}}_{t_i} + \sqrt{\Delta t} \, g(t_i) \, z_{t_i} \tag{53}$$

$$\mathbf{x}_{t_i} = \hat{\mathbf{x}}_{t_i}^{\text{noise}} + \Delta t \, g^2(t_i) \, s_\theta(\hat{\mathbf{x}}_{t_i}^{\text{noise}}, t_i), \tag{54}$$

where $z_{t_i} \sim \mathcal{N}(0, I)$. If the physics system is deterministic and $\Delta t$ is small enough, then we can approximate $\mathbf{x}_{t_i} \approx \hat{\mathbf{x}}_{t_i}$ and for the moment, we assume that we can write

$$\hat{\mathbf{x}}_{t_i}^{\text{noise}} = \mathbf{x}_{t_i} + \sqrt{\Delta t} \, g(t_i) \, z_{t_i}. \tag{55}$$

In this case, we can rewrite the 1-step loss $\mathcal{L}_{\text{single}}(\theta)$ from (14) to obtain the denoising score matching loss

$$\mathcal{L}_{\text{DSM}}(\theta) := \mathbb{E}_{(\mathbf{x}_{t_i}, \mathbf{x}_{t_{i+1}})} \left[ ||\mathbf{x}_{t_i} - \hat{\mathbf{x}}_{t_i}^{\text{noise}} - \Delta t \, g^2(t_i) \, s_\theta(\hat{\mathbf{x}}_{t_i}^{\text{noise}}, t_i)||_2^2 \right], \tag{56}$$

which is the same as minimizing

$$\mathbb{E}_{(\mathbf{x}_{t_i}, \mathbf{x}_{t_{i+1}})} \left[ ||s_\theta(\hat{\mathbf{x}}_{t_i}^{\text{noise}}, t_i) - \frac{1}{\Delta t \, g^2(t_i)} (\mathbf{x}_{t_i} - \hat{\mathbf{x}}_{t_i}^{\text{noise}})||_2^2 \right]. \tag{57}$$

Now, the idea presented in Vincent [Vin11] is that for score matching, we can consider a joint distribution $p_{t_i}(\mathbf{x}_{t_i}, \tilde{\mathbf{x}}_{t_i})$ of sample $\mathbf{x}_{t_i}$ and corrupted sample $\tilde{\mathbf{x}}_{t_i}$. Using Bayes' rule, we can write $p_{t_i}(\mathbf{x}_{t_i}, \tilde{\mathbf{x}}_{t_i}) = p_\sigma(\tilde{\mathbf{x}}_{t_i} | \mathbf{x}_{t_i}) p_{t_i}(\mathbf{x}_{t_i})$. The conditional distribution $p_\sigma(\cdot | \mathbf{x}_{t_i})$ for the corrupted sample is then modeled by a Gaussian with standard deviation $\sigma = \sqrt{\Delta t}\, g(t_i)$, i.e., we can write $\tilde{\mathbf{x}} = \mathbf{x} + \sqrt{\Delta t}\, g(t_i)\, z$ for $z \sim \mathcal{N}(0, I)$ similar to equation (55). Moreover, we can define the distribution of corrupted data $q_\sigma$ as

$$q_\sigma(\tilde{\mathbf{x}}) = \int p_\sigma(\tilde{\mathbf{x}}|\mathbf{x}) p_{t_i}(\mathbf{x}) d\mathbf{x}. \tag{58}$$

If $\sigma$ is small, then $q_\sigma \approx p_{t_i}$ and $\mathrm{KL}(q_\sigma \| p_{t_i}) \to 0$ as $\sigma \to 0$. Importantly, in this case, we can directly compute the score for $p_\sigma(\cdot | \mathbf{x})$ as

$$\nabla_{\tilde{\mathbf{x}}} \log p_\sigma(\tilde{\mathbf{x}} | \mathbf{x}) = \frac{1}{\sigma^2}(\mathbf{x} - \tilde{\mathbf{x}}). \tag{59}$$

Moreover, the theorem proven by Vincent [Vin11] means that we can use the score of the conditional distribution $p_\sigma(\cdot | \mathbf{x})$ to train $s_\theta(\mathbf{x}, t)$ to learn the score of $q_\sigma(\mathbf{x})$, i.e.

$$\arg\min_\theta \mathbb{E}_{\tilde{\mathbf{x}} \sim q_\theta} \left[ \| s_\theta(\mathbf{x}, t_i) - \nabla_{\tilde{\mathbf{x}}} \log q_\sigma(\tilde{\mathbf{x}}) \|_2^2 \right] \tag{60}$$

$$= \quad \arg\min_\theta \mathbb{E}_{\mathbf{x} \sim p_{t_i}, \tilde{\mathbf{x}} \sim p_\sigma(\cdot | \mathbf{x})} \left[ \| s_\theta(\mathbf{x}, t_i) - \nabla_{\tilde{\mathbf{x}}} \log p_\sigma(\tilde{\mathbf{x}} | \mathbf{x}) \|_2^2 \right]. \tag{61}$$

By combining (61) and (59), this exactly equals the denoising loss $\mathcal{L}_{\mathrm{DSM}}(\theta)$ in (57). As $q_\sigma \approx p_{t_i}$, we also obtain that $\nabla_{\mathbf{x}} \log q_\sigma(\mathbf{x}) \approx \nabla_{\mathbf{x}} \log p_{t_i}(\mathbf{x})$, so the network $s_\theta(\mathbf{x}, t_i)$ approximately learns the correct score for $p_{t_i}$.

We have assumed (55) that the only corruption for $\hat{\mathbf{x}}_{t_i}^{\mathrm{noise}}$ is the Gaussian noise. This is not true, as we have

$$\hat{\mathbf{x}}_{t_i}^{\mathrm{noise}} = \mathbf{x}_{t_i} + \sqrt{\Delta t}\, g(t_i)\, z_{t_i} + (\mathbf{x}_{t_{i+1}} - \Delta t\, \mathcal{P}(\mathbf{x}_{t_{i+1}}) - \mathbf{x}_{t_i}), \tag{62}$$

so there is an additional source of corruption, which comes from the numerical errors due to the term $\mathbf{x}_{t_{i+1}} - \Delta t\, \mathcal{P}(\mathbf{x}_{t_{i+1}}) - \mathbf{x}_{t_i}$. The conditional distribution $p_\sigma(\cdot | \mathbf{x})$ is only approximately Gaussian. Ideally, the effects of numerical errors are dominated by the Gaussian random noise. However, even small errors may accumulate for longer sequences of inference steps. To account for this, we argue that the multi-step loss $\mathcal{L}_{\mathrm{multi}}(\theta)$ should be used. During training, with the separation of physics update and denoising, the simulation state is first progressed from time $t_{i+1}$ to time $t_i$ using the reverse physics solver. This only yields a perturbed version of the simulation at time $t_i$ due to numerical inaccuracies. Then a small Gaussian noise is added and, via the denoising network $s_\theta(\mathbf{x}, t)$, the simulation state is projected back to the distribution $p_{t_i}$, which should also resolve the numerical errors. This is iterated, as discussed in section 2 in the main paper, depending on the sliding window size and location.

# B  Architectures

**ResNet**  We employ a simple ResNet-like architecture, which is used for the score function $s_\theta(\mathbf{x}, t)$ and the convolutional neural network baseline (ResNet) for the stochastic heat equation in section 3.2 as well as in section 3.4 again for the score $s_\theta(\mathbf{x}, t)$.

For experiments with periodic boundary conditions, we apply periodic padding with length 16, i.e., if the underlying 2-dimensional data dimensions are $N \times N$, the dimensions after the periodic padding are $(N + 16) \times (N + 16)$. We implement the periodic padding by tiling the input three times in $x$- and $y$-direction and then cropping to the correct sizes. The time $t$ is concatenated as an additional constant channel to the 2-dimensional input data when this architecture represents the score $s_\theta(\mathbf{x}, t)$.

The encoder part of our network begins with a single 2D-convolution encoding layer with 32 filters, kernel size 4, and no activation function. This is followed by four consecutive residual blocks, each consisting of 2D-convolution, LeakyReLU, 2D-convolution, and Leaky ReLU. All 2D convolutions have 32 filters with kernel size four and stride 1. The encoder part ends with a single 2D convolution with one filter, kernel size 1, and no activation. Then, in the decoder part, we begin with a transposed 2D convolution, 32 filters, and kernel size 4. Afterward, there are four consecutive residual blocks, analogous to the residual encoder blocks, but with the 2D convolution replaced by a transposed 2D convolution. Finally, there is a final 2D convolution with one filter and kernel size of 5. Parameter counts of this model and other models are given in table 2.

**UNet**  We use the UNet architecture with spatial dropout as described in [Mue+22], Appendix A.1, for the Bayesian neural network baseline of the stochastic heat equation experiment in section 3.2. The dropout rate is set to $0.25$. We do not include batch normalization and apply the same periodic padding as done for our ResNet architecture.

**FNO**  The FNO-2D architecture introduced in [Li+21] with $k_{\max,j} = 12$ Fourier modes per channel is used as a baseline for the stochastic heat equation experiment in section 3.2 and the learned physics surrogate model in section 3.4.

**Dil-ResNet**  The Dil-ResNet architecture is described in [Sta+21], Appendix A. This architecture represents the score $s_\theta(\mathbf{x}, t)$ in the buoyancy-driven flow with obstacles experiment in section 3.3. We concatenate the constant time channel analogously to the ResNet architecture. Additionally, positional information is added to the network input by encoding the $x$-position and $y$-position inside the domain in two separate channels.

| Architecture | Parameters |
|---|---|
| ResNet | 330 754 |
| UNet | 706 035 |
| Dil-ResNet | 336 915 |
| FNO | 465 377 |

Table 2: Summary of architectures.

## C    1D Toy SDE

For the 1D toy SDE discussed in section 3.1, we consider the SDE given by

$$dx = -\left[\lambda_1 \cdot \text{sign}(x)x^2\right] dt + \lambda_2 dw, \tag{63}$$

with $\lambda_1 = 7$ and $\lambda_2 = 0.03$. The corresponding reverse-time SDE is

$$dx = -\left[\lambda_1 \cdot \text{sign}(x)x^2 - \lambda_2^2 \cdot \nabla_x \log p_t(x)\right] dt + \lambda_2 dw. \tag{64}$$

Throughout this experiment, $p_0$ is a categorical distribution, where we draw either $1$ or $-1$ with the same probability. In figure 8, we show trajectories from this SDE simulated with the Euler-Maruyama method. Trajectories start at $1$ or $-1$ and approach $0$ as $t$ increases.

**Neural network architecture**   We employ a neural network $s_\theta(x, t)$ parameterized by $\theta$ to approximate the score via the 1-step loss, the multi-step loss, implicit score matching [Hyv05, ISM] and sliced score matching with variance reduction [Son+19, SSM-VR]. In all cases, the neural network is a simple multilayer perceptron with elu activations and five hidden layers with 30, 30, 25, 20, and then 10 neurons for the last hidden layer.

We use the Adam optimizer with standard hyperparameters as described in the original paper [KB15]. The learning rate, batch size, and the number of epochs depend on the data set size (100% with 2 500 trajectories, 10%, or 1%) and are chosen to ensure convergence of the training loss.

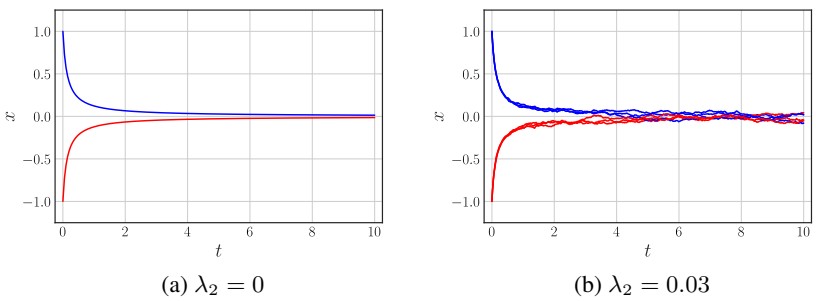

(a) $\lambda_2 = 0$                                  (b) $\lambda_2 = 0.03$

Figure 8: Trajectories from SDE (63) with $\lambda_2 = 0$ (a) and $\lambda_2 = 0.03$ (b).

**Training - 1-step loss**   For the 1-step loss and all data set sizes, we train for 250 epochs with a learning rate of 10e-3 and batch size of 256. In the first phase, we only keep every 5th point of a trajectory and discard the rest. Then, we again train for 250 epochs with the same batch size and a learning rate of 10e-4 but keep all points. Finally, we finetune the network with 750 training epochs and a learning rate of 10e-5.

**Training - multi-step loss**   For the multi-step loss and 100% of the data set, we first train with the 1-step loss, which resembles a sliding window size of 2. We initially train for 1 000 epochs with a batch size of 512 and a learning rate of 10e-3, where we keep only every 5th point on a trajectory and discard the rest. Then, with a decreased learning rate of 10e-4, we begin training with a sliding window size of $S = 2$ and increment it every 1 000 epochs by one until $S_{\max} = 10$. In this phase, we train on all points without any removals.

**Training - ISM**   For ISM, we compute the partial derivative $\partial s_\theta(\mathbf{x})_i / \partial \mathbf{x}_i$ using reverse-mode automatic differentiation in JAX (jax.jacrev). For 100% and 10% of the data set, we train for 2 000 epochs with a learning rate of 10e-3 and batch size of 10 000. Then we train for an additional 2 000 epochs with a learning rate 10e-4. For 1%, increase the number of epochs to 20 000.

**Training - SSM-VR**   For sliced score matching with variance reduction [Son+19, SSM-VR], we use the same training setup as for ISM.

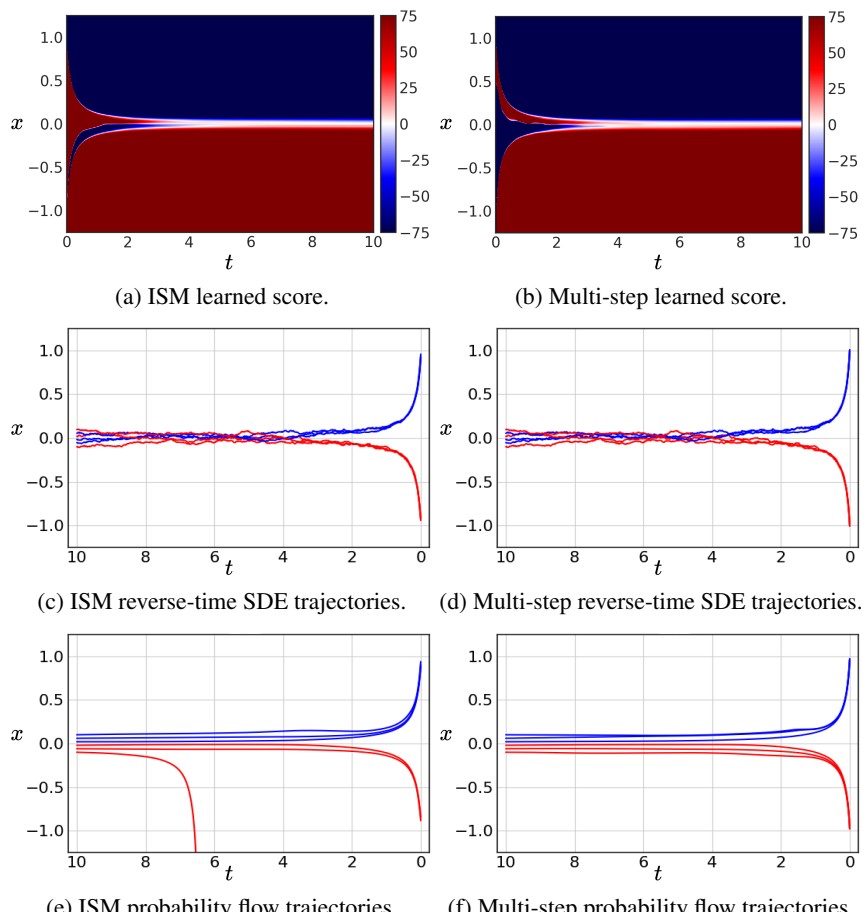

(a) ISM learned score.  (b) Multi-step learned score.

(c) ISM reverse-time SDE trajectories.  (d) Multi-step reverse-time SDE trajectories.

(e) ISM probability flow trajectories.  (f) Multi-step probability flow trajectories.

Figure 9: Comparison of Implicit Score Matching (ISM, left) and our proposed training with the multi-step loss (Multi-step, right). Colormap in (a) and (b) truncated to [-75, 75].

| Method | Probability flow ODE | | | Reverse-time SDE | | |
|---|---|---|---|---|---|---|
| | Data set size | | | Data set size | | |
| | 100% | 10% | 1% | 100% | 10% | 1% |
| multi-step | **0.97**±0.04 | **0.91**±0.05 | **0.81**±0.01 | **0.99**±0.01 | 0.94±0.02 | **0.85**±0.06 |
| 1-step | 0.78±0.16 | 0.44±0.13 | 0.41±0.13 | 0.93±0.05 | 0.71±0.10 | 0.75±0.10 |
| ISM | 0.19±0.05 | 0.15±0.15 | 0.01±0.01 | 0.92±0.05 | 0.94±0.01 | 0.52±0.22 |
| SSM-VR | 0.17±0.16 | 0.49±0.24 | 0.27±0.47 | 0.88±0.06 | 0.94±0.06 | 0.67±0.23 |

Table 3: Posterior metric $Q$ for 1 000 predicted trajectories averaged over three runs.

**Comparison**  We directly compare the learned score for the reverse-time SDE trajectories and the probability flow trajectories between ISM and the multi-step loss in figure 9 trained on the full data set. The learned score of ISM and the multi-step loss in figure 9a and figure 9b are visually very similar, showing that our method and loss learn the correct score. Overall, after finetuning both ISM and the multi-step loss, the trajectories of the multi-step loss are more accurate compared to ISM. For example, in figure 9e, a trajectory explodes to negative infinity. Also, trajectories from the multi-step loss end in either $-1$ or $1$, while ISM trajectories are attenuated and do not fully reach $-1$ or $1$ exactly, particularly for the probability flow ODE.

**Results of Table 1 in Main Paper**  We include the standard deviations of table 1 from the main paper in table 3 above. The posterior metric $Q$ is very sensitive to the learned score $s_\theta(\mathbf{x}, t)$. Overall, our proposed multi-step loss gives the most consistent and reliable results.

**Empirical verification of Theorem 3.1** For the quadratic SDE equation 63, the analytic score is non-trivial, therefore a direct comparison of the learned network $s_\theta(\mathbf{x}, t)$ and the true score $\nabla_{\mathbf{x}} \log p_t(\mathbf{x})$ is difficult. However, we can consider the SDE with affine drift given by

$$dx = -\lambda x dx + g dW \tag{65}$$

with $\lambda = 0.5$ and $g \equiv 0.04$. Because this SDE is affine and there are only two starting points $-1$ and $1$, we can write the distribution of states starting in $x_0$ as a Gaussian with mean $\mu(t; x_0) = x_0 e^{-\lambda t}$ and variance $\sigma^2(t; x_0) = \frac{g^2}{2\lambda}(1 - e^{-2\lambda t})$, see [Øks03]. Then, the score at time $t$ and position $x$ conditioned on the starting point $x_0$ is $(x - \mu(t; x_0))/\sigma^2(t; x_0)$. See figure 10 for a visualization of the analytic score and a comparison with the learned score. The learned score from the 1-step training matches the analytic score very well in regions where the data density is sufficiently high.

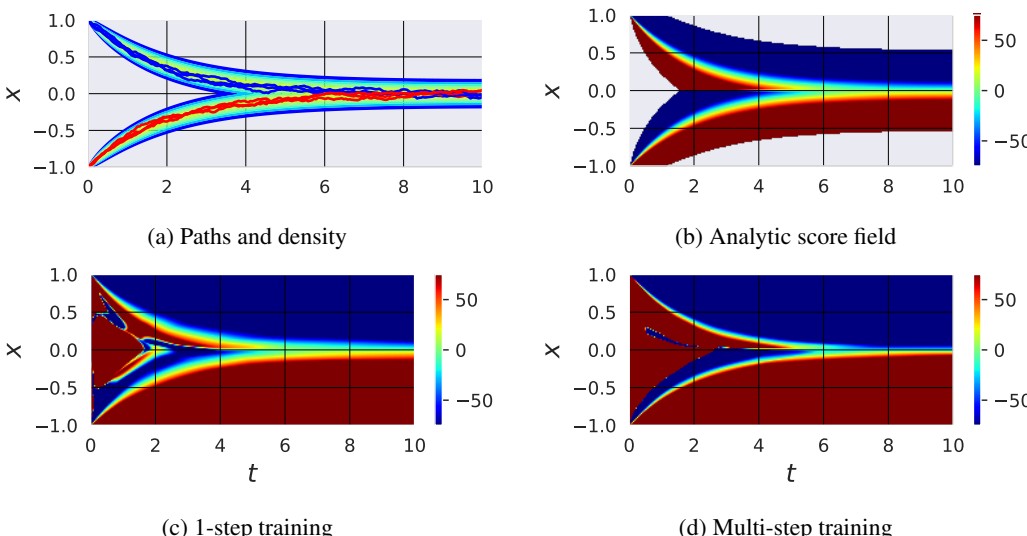

(a) Paths and density

(b) Analytic score field

(c) 1-step training

(d) Multi-step training

Figure 10: 1-step training score matches analytic score. (a) shows some paths sampled from SDE equation (65) and a contour of the density. (b) is the analytic score field, and (c) and (d) are visualizations of the learned score with 1-step and multi-step training. Scores from the multi-step training correspond to more narrow high-density regions. This implies that during inference, trajectories are pulled more strongly to modes of the training data set distribution than for the 1-step training.

# D Stochastic Heat Equation

**Spectral solver**   The physics of the 2-dimensional heat equation for $\mathbf{x} \in \mathbb{R}^{d \times d}$ can be computed analytically. The solver $\mathcal{P}_h^{\Delta t}(\mathbf{x})$ simulates the systems $\mathbf{x}$ forward in time by a fixed $\Delta t$ using the (shifted) Fourier transformation $\mathcal{F}$. In particular, we can implement the solver with

$$\mathcal{P}_h^{\Delta t}(\mathbf{x}) = \mathcal{F}^{-1}\left(A(\Delta t) \circ \mathcal{F}(\mathbf{x})\right), \tag{66}$$

where $\circ$ denotes element-wise multiplication and $A(\Delta t)_{ij} \in \mathbb{R}^{d \times d}$ is a matrix with entries $A(\Delta t)_{ij} := \exp\left(-\Delta t \cdot \min(i, j, d - i, j - i)\right)$. The power spectrum of $\mathbf{x}$ is scaled down by $A(\Delta t)$, and higher frequencies are suppressed more than lower frequencies (for $\Delta t > 0$). If noise is added to $\mathbf{x}$, then this means that especially higher frequencies are affected. Therefore the inverse transformation (when $\Delta t > 0$) exponentially scales contributions by the noise, causing significant distortions for the reconstruction of $\mathbf{x}$.

**Spectral loss**   We consider a spectral error based on the two-dimensional power spectral density. The radially averaged power spectra $s_1$ and $s_2$ for two images are computed as the absolute values of the 2D Fourier transform raised to the second power, which are then averaged based on their distance (in pixels) to the center of the shifted power spectrum. We define the spectral error as the weighted difference between the log of the spectral densities

$$\mathcal{L}(s_1, s_2) = \sum_k w_k |\log(s_{1,k}) - \log(s_{2,k})| \tag{67}$$

with a weighting vector $w \in \mathbb{R}^d$ and $w_k = 1$ for $k \leq 10$ and $w_k = 0$ otherwise.

**Training**   For inference, we consider the linear time discretization $t_n = n\Delta t$ with $\Delta t = 0.2/32$ and $t_{32} = 0.2$. During training, we sample a random time discretization $0 \leq t_0 < t_2' < .... < t_{31}' < t_{32}$ for each batch based on $t_n$ via $t_n' \sim \mathcal{U}(t_n - \Delta t/2, t_n + \Delta t/2)$ for $n = 1, ..., 31$ and adjust the reverse physics step based on the time difference $t_i - t_{i-1}$. In the first training phase, we consider the multi-step loss with a sliding window size of $S = 6, 8, ..., 32$ steps, where we increase $S$ every two epochs. We use Adam to update the weights $\theta$ with learning rate $10^{-4}$. We finetune the network weights for $80$ epochs with an initial learning rate of $10^{-4}$, which we reduce by a factor of $0.5$ every $20$ epochs.

**$s_\theta$ only version**   For the 1-step loss, this method is similar to Rissanen et al. [RHS22], which proposes a classical diffusion-like model that generates data from the dynamics of the heat equation. Nonetheless, the implementation details and methodology are analogous to the multi-step loss training, except that the reverse physics step $\tilde{\mathcal{P}}^{-1}$ is not explicitly defined but instead learned by the network $s_\theta(\mathbf{x}, t)$ together with the score at the same time. We make use of the same ResNet architecture as the default variant. Except for the reverse physics step, the training setup is identical. Although the network $s_\theta(\mathbf{x}, t)$ is not trained with any noise for a larger sliding window with the multi-step loss, we add noise to the simulation states for the SDE inference, while there is no noise for the ODE inference.

**Baseline methods**   All other baseline methods are trained for $80$ epochs using the Adam optimizer algorithm with an initial learning rate of $10^{-4}$, which is decreased by a factor of $0.5$ every $20$ epochs. For the training data, we consider solutions to the heat equation consisting of initial state $\mathbf{x}_0$ and end state $\mathbf{x}_T$.

**Test-time distribution shifts**   We have tested the effects of test-time distribution shifts for the heat equation experiment. We train the score network $s_\theta$ for a specific combination of diffusivity $\alpha$ and noise $g$ and vary both parameters for testing. We modify both the simulator and test ground truth for the updated diffusivity and noise. See figure 11. Overall, for small changes of the parameters, there is very little overfitting. Changes in the reconstruction MSE and spectral error can mainly be explained by making the task itself easier or harder to which our network generalizes nicely, e.g., less noise or higher diffusivity leads to smaller reconstruction error.

**Additional results**   We show visualizations for the predictions of different methods in figure 12 and figure 13.

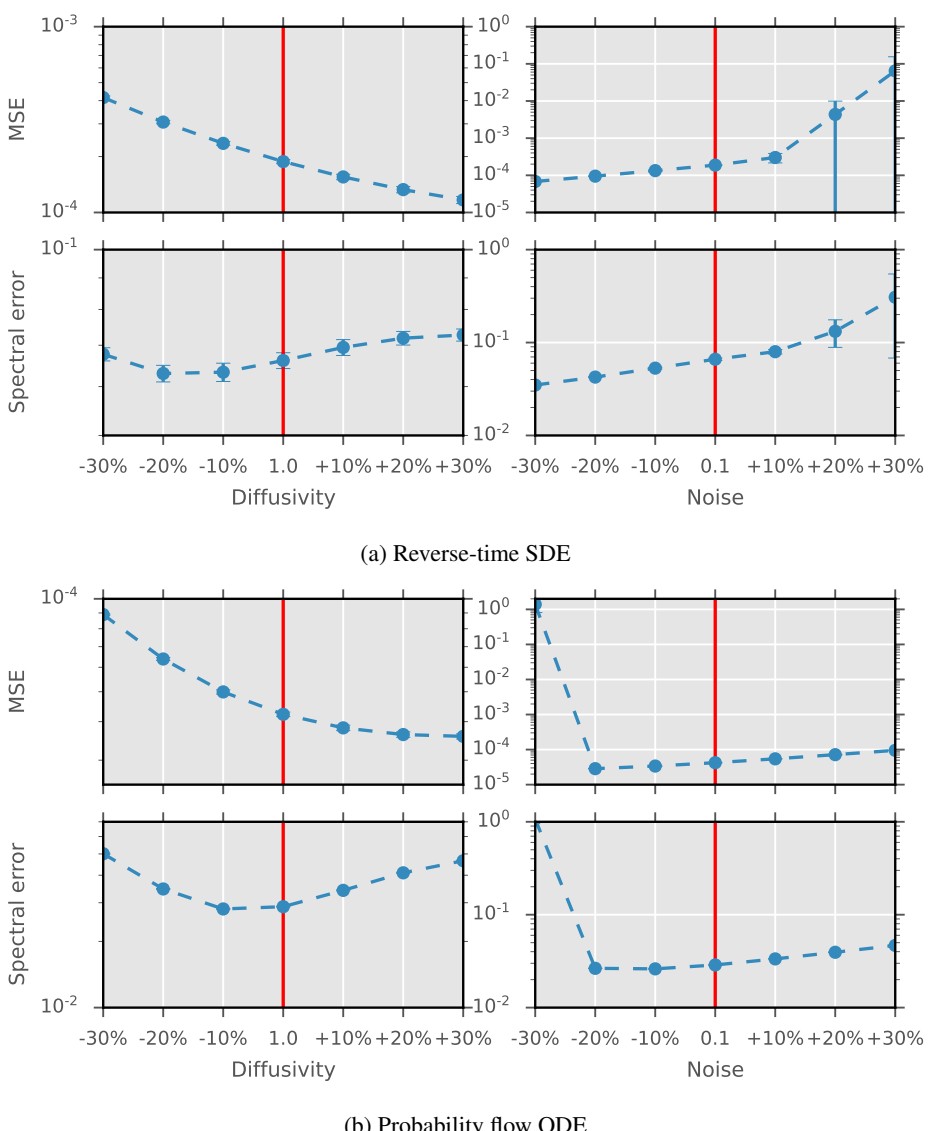

(a) Reverse-time SDE

(b) Probability flow ODE

Figure 11: Test-time distribution shifts for noise and diffusivity. The correction network $s_\theta$ is trained on diffusivity $\alpha = 1.0$ and noise $g \equiv 0.1$. During testing, we vary the diffusivity of the test data set and simulator as well as the noise for inference. Especially the low spectral errors indicate that the network generalizes well to the new behavior of the physics.

# E  Buoyancy-driven Flow with Obstacles

We use semi-Lagrangian advection for the velocity and MacCormack advection for the hot marker density within a fixed domain $\Omega \subset [0, 1] \times [0, 1]$. The temperature dynamics of the marker field are modeled with a Boussinesq approximation.

**Training**   We train all networks with Adam and learning rate $10^{-4}$ with batch size 16. We begin training with a sliding window size of $S = 2$, which we increase every 30 epochs by 2 until $S_{\mathrm{max}} = 20$.

**Separate vs. joint updates**   We compare a joint update of the reverse physics step and corrector function $s_\theta$, see figure 7b, and a separate update of reverse physics step and corrector function, see figure 7c. An evaluation regarding the reconstruction MSE and perceptual distance is shown in

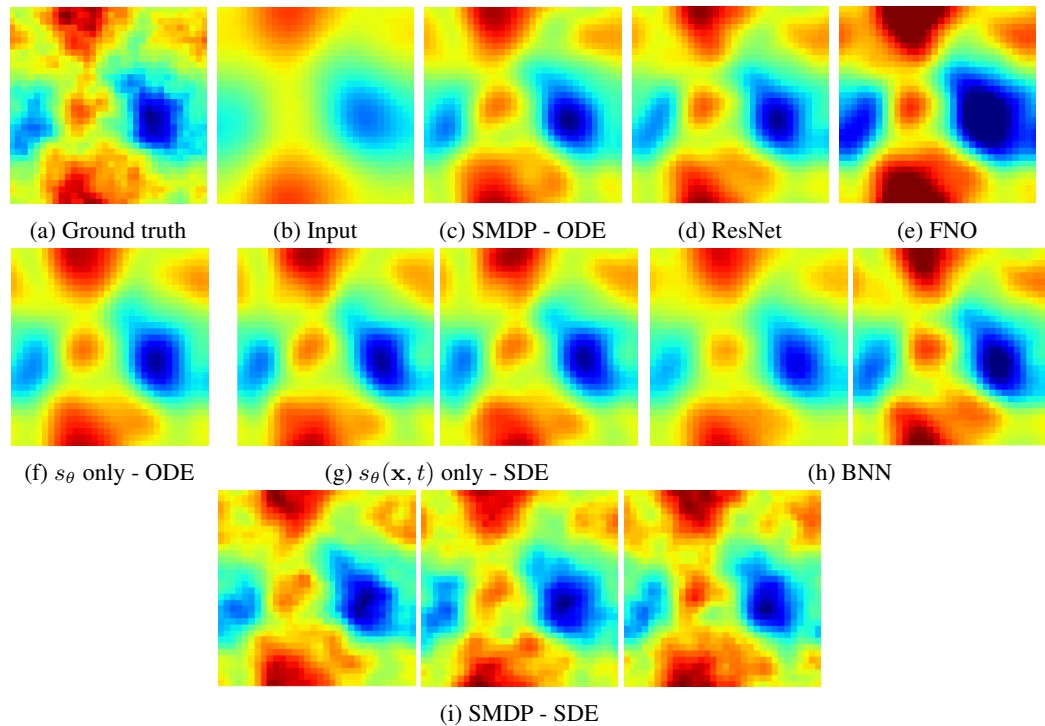

Figure 12: Predictions of different methods for the heat equation problem (example 1 of 2).

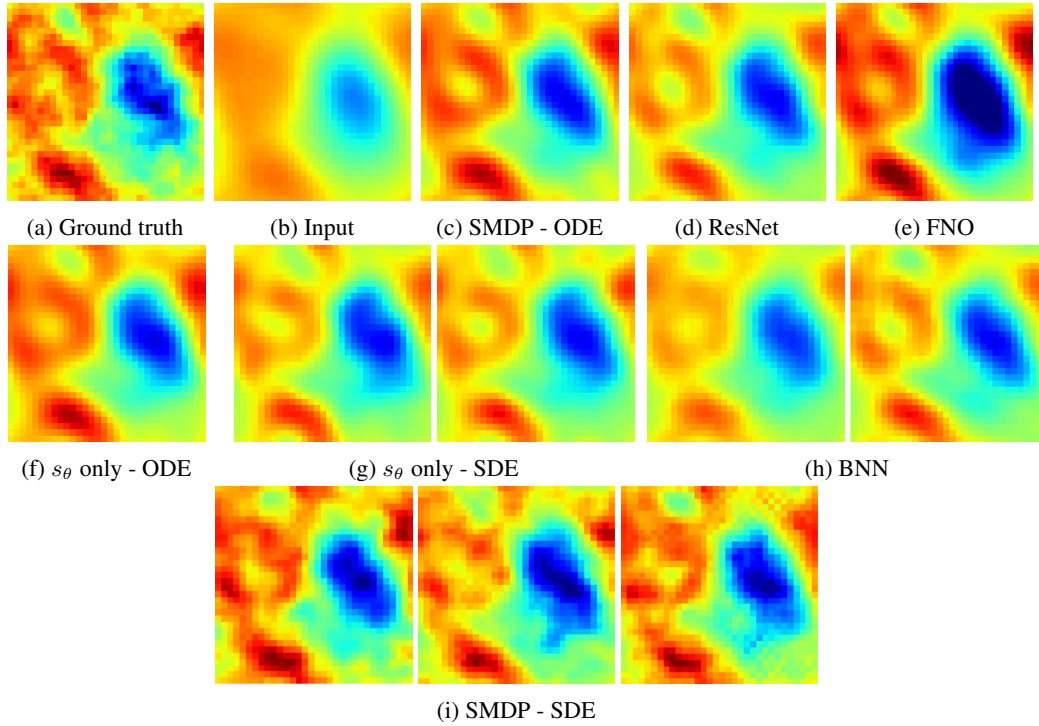

Figure 13: Predictions of different methods for the heat equation problem (example 2 of 2). Neither the BNN nor the "$s_\theta$ only" model can produce small-scale structures.

figure 14. Both training and inference variants achieve advantages over "$\tilde{\mathcal{P}}^{-1}$ only" and "$s_\theta$ only"

approaches. Overall, there are no apparent differences for the ODE inference performance but slight benefits for the SDE inference when separating physics and corrector update.

**Limited-memory BFGS**  We use numerical optimization of the marker and velocity fields at $t = 0.35$ to match the target smoke and velocity fields at $t = 0.65$ using limited-memory BFGS [LN89, LBFGS] and the differentiable forward simulation implemented in *phiflow* [Hol+20]. Our implementation directly uses the LBFGS implementation provided by *torch.optim.LBFGS* [Pas+19]. As arguments, we use $history\_size = 10$, $max\_iter = 4$ and $line\_search\_fn = strong\_wolfe$. Otherwise, we leave all other arguments to the default values. We optimize for 10 steps which takes ca. 240 seconds per sample on a single NVIDIA RTX 2070 gpu.

**Diffusion posterior sampling DPS**  An additional baseline for this problem is diffusion posterior sampling for general noisy inverse problems [Chu+23, DPS]. As a first step, we pretrain a diffusion model on the data set of marker and velocity fields at $t = 0.35$. We use the mask for obstacle positions as an additional conditioning input to the network to which no noise is applied. Our architecture and training closely resemble Denoising Diffusion Probabilistic Models [HJA20, DDPM]. Our network consists of ca. 18.44 million parameters trained for 100k steps and learning rate $1 \times 10^{-4}$ using cosine annealing with warm restarts ($T_0 = 50000$, $\eta_{min} = 5 \times 10^{-6}$). The measurement operator $\mathcal{A}$ is implemented using our differentiable forward simulation. We consider the Gaussian version of DPS, i.e., Algorithm 1 in [Chu+23] with $N = 1000$. We fix the step size $\zeta_i$ at each iteration $i$ to $1/||y - \mathcal{A}(\hat{\mathbf{x}}_0(\mathbf{x}_i))||$. For each inference step, we are required to backpropagate gradients through the diffusion model and the forward simulation. Inference for a single sample requires ca. 5000 seconds on a single NVIDIA RTX 2070 gpu.

**Additional results**  We provide more detailed visualizations for the buoyancy-driven flow case in figure 16 and figure 17. These again highlight the difficulties of the reverse physics simulator to recover the initial states by itself. Including the learned corrections significantly improves this behavior.

In figure 15, we also show an example of the posterior sampling for the SDE. It becomes apparent that the inferred small-scale structures of the different samples change. However, in contrast to cases like the heat diffusion example, the physics simulation in this scenario leaves only little room for substantial changes in the states.

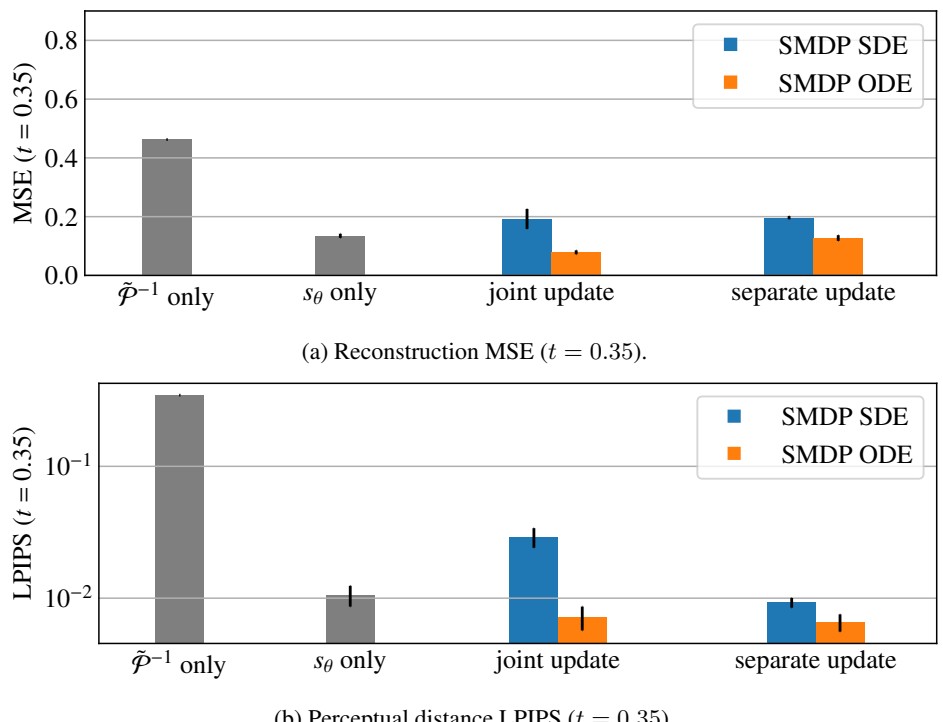

(a) Reconstruction MSE ($t = 0.35$).

(b) Perceptual distance LPIPS ($t = 0.35$).

Figure 14: Comparison of separate and joint updates averaged over three runs.

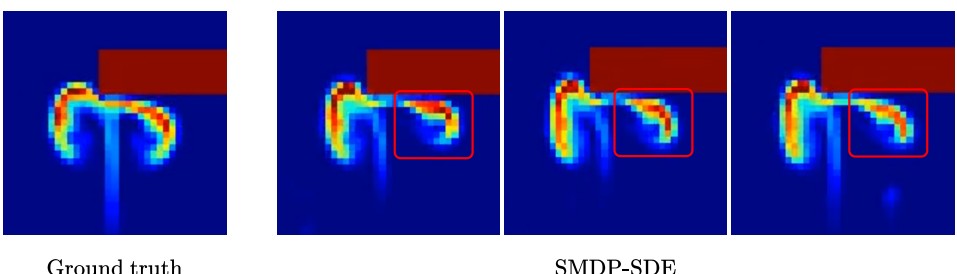

Ground truth                                    SMDP-SDE

Figure 15: Comparison of SMDP-SDE predictions and ground truth for buoyancy-driven flow at $t = 0.36$.

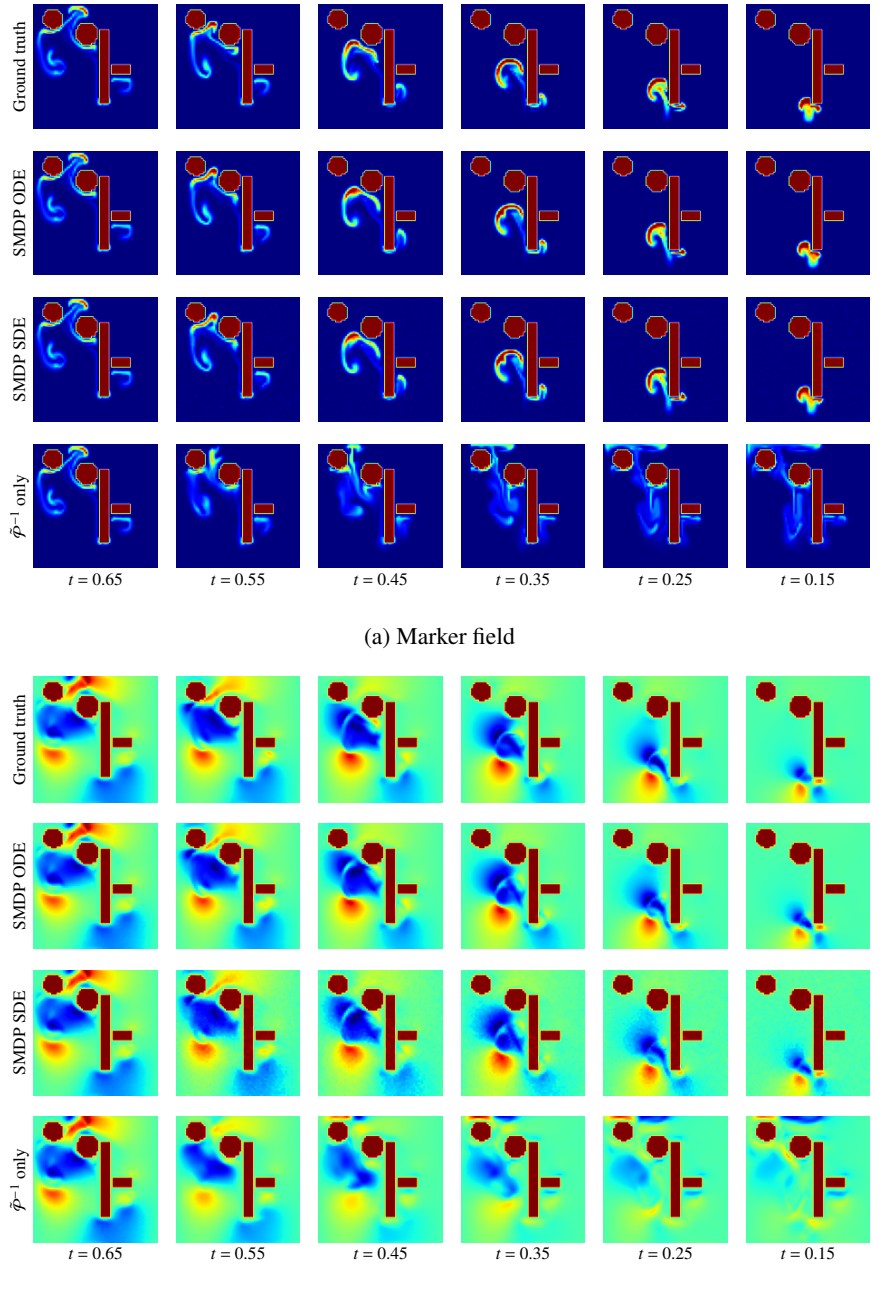

(a) Marker field

(b) Velocity field ($x$-direction)

Figure 16: Predictions for buoyancy-driven flow with obstacles (example 1 of 2).

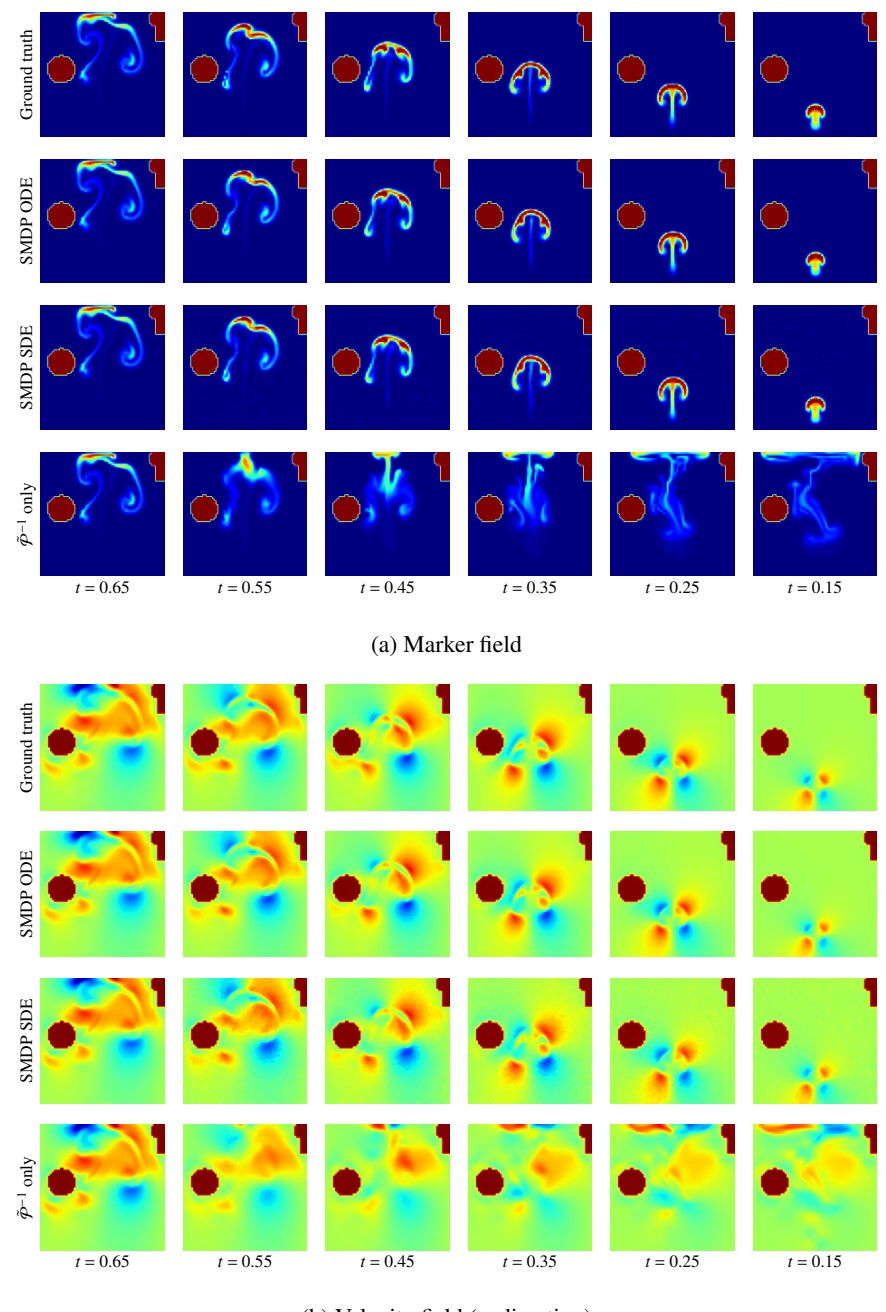

(a) Marker field

(b) Velocity field ($x$-direction)

Figure 17: Predictions for buoyancy-driven flow with obstacles (example 2 of 2).

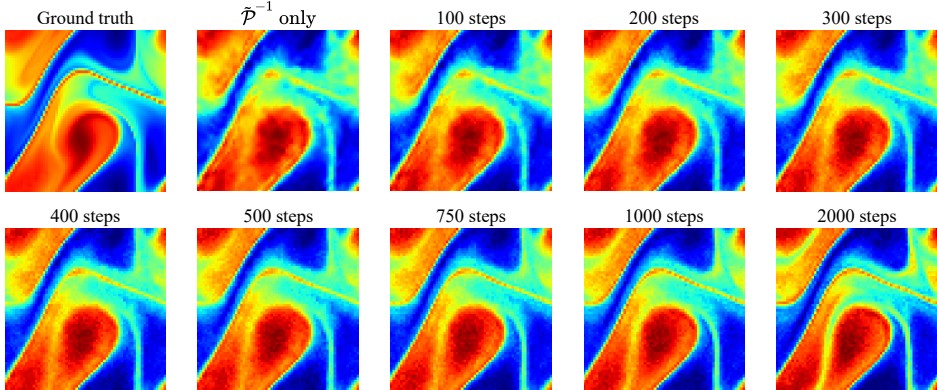

Figure 18: Steps of Langevin dynamics for $\epsilon = 2 \times 10^{-5}$.

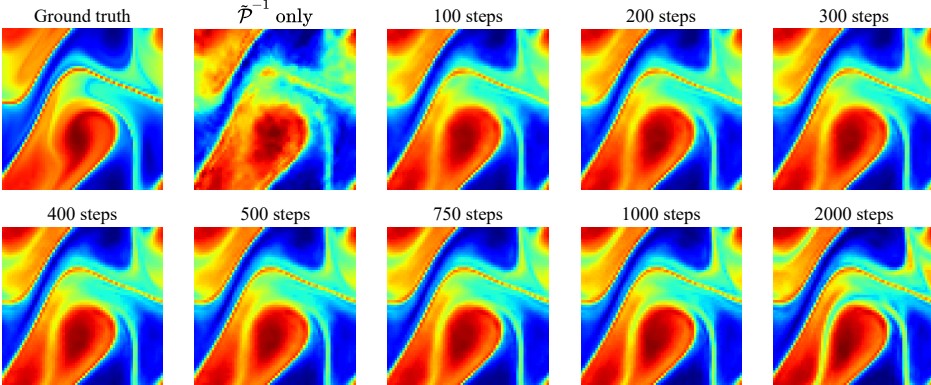

Figure 19: Steps with Langevin dynamics for $\epsilon = 2 \times 10^{-5}$ and an additional step with $\Delta t \, s_\theta(\mathbf{x}, t)$ which smoothes the images.

## F  Isotropic turbulence

**Training**   For the physics surrogate model $\tilde{\mathcal{P}}^{-1}$, we employ an FNO neural network, see appendix B, with batch size 20. We train the FNO for 500 epochs using Adam optimizer with learning rate $10^{-3}$, which we decrease every 100 epochs by a factor of 0.5. We train $s_\theta(\mathbf{x}, t)$ with the ResNet architecture, see appendix B, for 250 epochs with learning rate $10^{-4}$, decreased every 50 epochs by a factor of 0.5 and batch size 6.

**Refinement with Langevin Dynamics**   Since the trained network $s_\theta(\mathbf{x}, t)$ approximates the score $\nabla_\mathbf{x} \log p_t(\mathbf{x})$, it can be used for post-processing strategies [WT11; SE19]. We do a fixed point iteration at a single point in time based on Langevin Dynamics via:

$$\mathbf{x}_t^{i+1} = \mathbf{x}_t^i + \epsilon \cdot \nabla_\mathbf{x} \log p_t(\mathbf{x}_t^i) + \sqrt{2\epsilon} z_t \tag{68}$$

for a number of steps $T$ and $\epsilon = 2 \times 10^{-5}$, cf. figure 18 and figure 19. For a prior distribution $\pi_t$, $\mathbf{x}_t^0 \sim \pi_t$ and by iterating (68), the distribution of $\mathbf{x}_t^T$ equals $p_t$ for $\epsilon \to 0$ and $T \to \infty$. There are some theoretical caveats, i.e., a Metropolis-Hastings update needs to be added in (68), and there are additional regularity conditions [SE19].

**Additional results**   We show additional visualizations of the ground truth and reconstructed trajectories in figure 20 and figure 21.

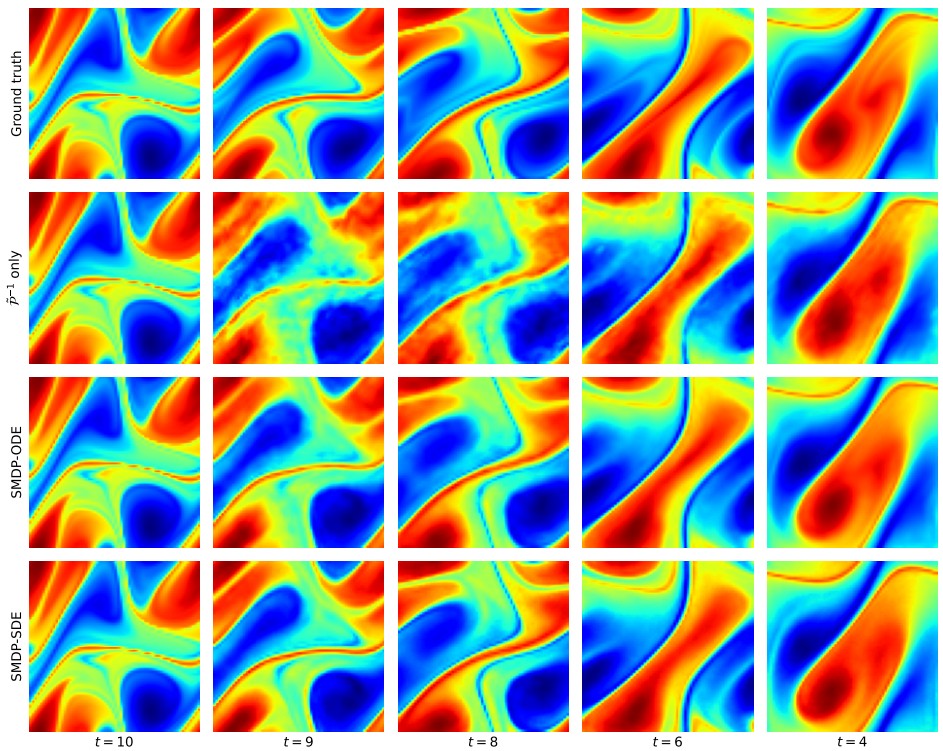

Figure 20: Predictions for isotropic turbulence (example 1 of 2).

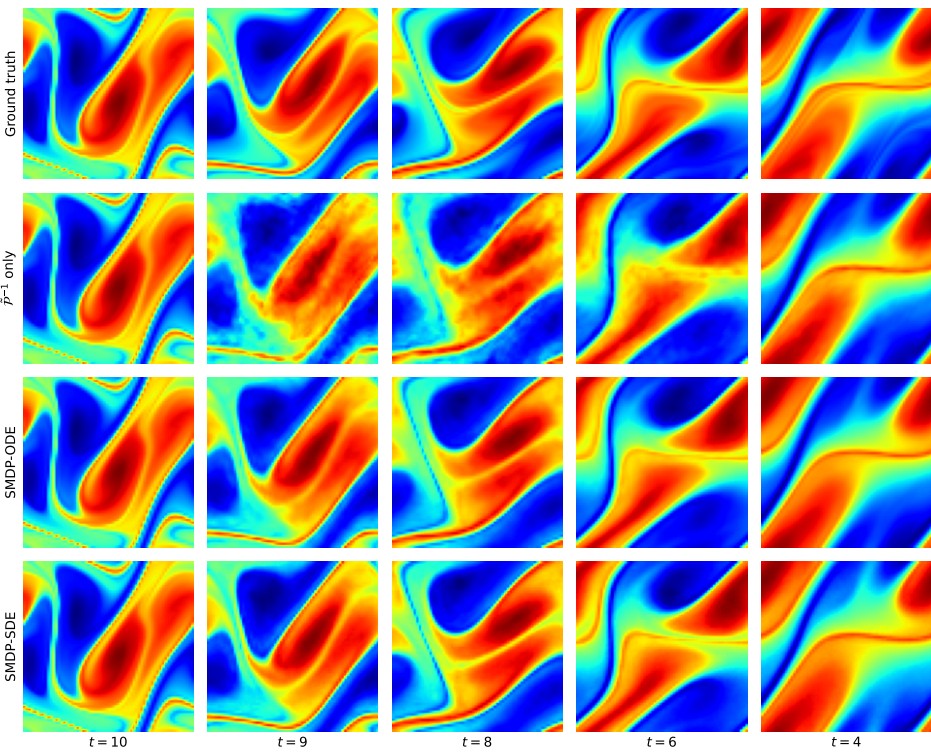

Figure 21: Predictions for isotropic turbulence (example 2 of 2).

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
