# OpenReview forum: "Solving Inverse Physics Problems with Score Matching"
_NeurIPS.cc/2023/Conference — NeurIPS 2023 poster_

### Official Review · Reviewer_3QqF · 2023-07-01

**Soundness:** 4 excellent
**Presentation:** 3 good
**Contribution:** 4 excellent
**Rating:** 7
**Confidence:** 4

**Summary:**

The authors leverage the framework of score matching, which has become popular for training diffusion-based models for generative tasks, to reverse physical processes defined by forward stochastic differential equations (SDEs). Given a system state at time t=T, they propose to solve for the initial conditions at t=0 by iteratively applying a reverse-time diffusion process defined by a reverse physics simulator, a diffusion term, and the score of the data distribution. In the main contribution of the paper, the authors introduce both a 1-step and multi-step loss for training a network to learn the diffusion and score terms for solving the reverse SDE. They prove the equivalence of their proposed training objectives to vanilla denoising score matching (and a related variational objective in the multi-step case). Extensive experiments demonstrate the efficacy of their method compared to baselines, and ablation studies show the utility of the proposed multi-step objective.

**Strengths:**

### Originality

To my knowledge, this is the first work that considers the drift term in the typical forward SDE used in diffusion-based generative models to be a realistic physical process. Other works [1,2] have considered different diffusion processes besides additive Gaussian noise, but even these use artificial processes such as synthetic blur and pixel masking. I appreciate that the authors of the current paper identify the connection between physically-defined differential equations and those used in training score-based models, and propose a novel method for solving physics problems.

### Quality

- The authors perform extensive experiments grounded in real-world physical processes and show the effects of varying numerous design choices in each experiment
- The proposed approach demonstrates superior performance compared to baselines across various settings
- Ablation studies show the utility of the multi-step objective over the 1-step variant
- The figures are well-made, particularly Figure 1 which clearly lays out the key ideas of the proposed approach

### Clarity

- The introduction does a good job of laying out the current state of diffusion-based generative models, the connection to physics processes, and the contribution of the current work
- Clear descriptions of the problem setup, model training and inference, and hyperparameters are given for each experiment

### Significance

Physics-based inverse problems arise in many fields such as astronomy, geophysics, and wireless communication, so finding better solutions is a highly significant problem with broad interest. Furthermore, quantifying the uncertainty of these solutions is crucial to the downstream decision-making process. The authors of the current paper provide a principled approach to both solve and provide uncertainty estimates (using multiple samples from the SDE) for inverse problems in physics.

### References:

[1] G. Daras, M. Delbracio, H. Talebi, A. Dimakis, and P. Milanfar, “Soft Diffusion: Score Matching with General Corruptions,” Transactions on Machine Learning Research, 2023, [Online].

[2] A. Bansal et al., Cold Diffusion: Inverting Arbitrary Image Transforms Without Noise. 2022.

**Weaknesses:**

- I believe that the loss in Eq (2) is incorrect. The quantity within the squared L2 norm should be $x_m$ minus the quantity on the right-hand side of Eq (1). In its current form, Eq (2) does not match Eq (3) when Eq (3) is expanded with window S=2.
- While the thorough experimental details are appreciated, I believe some of that can be pushed to the appendix. More space should be dedicated to expanding on motivation and design choices. As it currently stands, section 2 reads as a constant flow of information with insufficient context for the proposed objectives.
- I believe that the organization of the sections could be improved by introducing the typical score matching SDE subject matter first, identifying the differences between that formulation and the physics-based inverse problem formulation, then motivating your proposed approach within this context.
- An obvious concern to me regarding the multi-step training is the huge memory expense arising from the recursive calls to $s_{\theta}$. However, the authors did not address this point in the main paper.
- The authors state that, for decreasing time step $\Delta t$, the reverse physics simulator is equivalent to the negative of the forward simulator (line 113). However, it is not clarified whether this is a simplifying assumption, true in general, or true in the specific case of the reverse-time ODE. The authors expand on the specific choices of the reverse simulators in the experiments section, but the relationship between the forward and reverse simulators remains unclear to me.


**Questions:**

- What is your reasoning for training your model with the 1-step loss between $x_m$ and $x_{m-1}$ instead of $x_m$ and $x_0$ (as in typical diffusion-based generative models)? As I understand it, the forward and reverse physical processes are not time-dependent, and arbitrarily large time steps can be taken by simply increasing $\Delta t$. Intuitively, this method makes more sense to me than your proposed approach, and would also remove the need for multi-step training.
- How did you deal with the memory expense during multi-step training? Did you find that you needed to use smaller data and network sizes to fit the gradients in memory, or were there tricks and optimizations you used to reduce memory use?
- One concern is that your model will overfit to the physical parameters at training and perform poorly if there is a test-time distribution shift. How robust is your proposed approach to test-time shifts in the SDE parameters (namely, the coefficients of the simulator and diffusion terms)? I understand that there is limited time for responses, but a small experiment would be appreciated and may convince me to raise my score.

**Limitations:**

The authors mention limitations in their conclusion.

---

> ### Author Rebuttal · Authors · 2023-08-10
>
> We thank the reviewer for their valuable review and helpful suggestions.
>
> - We restructured our paper based on the feedback we received. In particular, we moved the related work section forward so that it follows the introduction and moved parts of the experimental details to the appendix to improve overall readability.
>
> - **Memory expense**:  Possible solutions to save memory are gradient cutting and gradient checkpointing. We have tested the effects of gradient cutting for the heat equation task. In this experiment, we unrolled the entire simulation with the multi-step loss, but stop backpropagating the gradient after n steps. n = 32 corresponds the entire simulation trajectory. Our results show that there are no more performance benefits when backpropagating the gradient for more than 8 simulation steps for the ODE inference.
> | Gradient cutting after | Avg. reconstruction MSE  |
> | ----| ----------- |
> | 1   |  4.6e-5   |
> | 2   | 1.4e-5    |
> | 4   | 1.0e-5    |
> | 8   | 0.8e-5    |
> | 16 | 0.8e-5  |
> | 32 | 0.8e-5  |
>
> - For the evaluation of the tasks in the paper, we trained with no specific optimizations of the memory consumption. The most expensive task in terms of memory is the buoyancy-driven flow with obstacles experiment for which we used a single A100 GPU with 40GB of memory. Our neural networks are very small compared to architectures for diffusion models and generative modeling and have a size that’s comparable to typical learned correction approaches, cf. e.g., [Um et al. 2020, Kochkov et al. 2021].
>
> -  **Reverse-physics simulator**: In general there is no reverse simulator for larger time scales, as information can be lost and the initial state might not be possible to reconstruct. If we solve the underlying PDE iteratively this can be described by the update rule $x_{t+1} = x_{t} + \Delta t$ PDEupdate$(x_t)$. Then, we may approximate $x_{t} \approx x_{t+1} - \Delta t$ PDEupdate$(x_{t+1})$. So locally we can identify the reverse simulator with the negative of the forward simulator. In our experiments with non-learned simulators, we use the implementation of the forward simulator to obtain the reverse simulator by using a negative step size $\Delta t$.  We included additional comments about this in the main paper as well as expanding upon the relationship in the appendix.
>
> -  **1-step loss between $\mathbf{x}_0$ and $\mathbf{x}_m$**: In theory it would be possible to consider the 1-step loss between $\mathbf{x}_0$ and $\mathbf{x}_m$ and compute the trajectory in a single step. However, in practice there are numerical issues that need to be considered. In standard diffusion models, the score is easy to compute given sample and noise, but in our case, we still require an ODE solver. Our method is described from the viewpoint of Euler steps. However, other methods for time integration could be considered, which we leave to future work. Importantly, there are considerations regarding the implementation of the reverse-physics simulator. We found that many small steps + correction for each step is numerically significantly more stable than a single big step with one correction only.
>
> - **Test-time distribution shifts**: We have tested the effects of test-time distribution shifts for the heat equation experiment. Here we train the score network for a specific combination of diffusivity and noise and vary both parameters for testing (always updating both the simulator and test ground truth), see the pdf in the global response. Overall, for small changes of the parameters, there seems to be very little overfitting. Changes in the reconstruction MSE and spectral error can mainly be explained by making the task itself easier/harder to which our network generalizes nicely (e.g. less noise or higher diffusivity -> smaller reconstruction error).

---

> > ### Comment · Reviewer_3QqF · 2023-08-16
> >
> > Thank you to the authors for your detailed responses as well as for your additional experiments. As I stated in my review, I will raise my score as I believe that you have sufficiently answered most of my questions.

---

### Official Review · Reviewer_mtJA · 2023-07-05

**Soundness:** 2 fair
**Presentation:** 3 good
**Contribution:** 2 fair
**Rating:** 4
**Confidence:** 3

**Summary:**

The paper proposes using score matching to learn the backward process of a given forward SDE, notably coming from a physics application. The paper states that this can be used to simulate backward the distribution of the initial condition given the end state, by starting from the end state and drawing trajectories backward according to the backward SDE. This can be of interest in situations where the physics of a system is not invertible, as the authors point out is the case for most of the macroscopic state governing equations.

The authors propose two inference procedures, a SDE (sampler) and an ODE (deterministic) and test in 4 different settings, with different levels of complexity.

**Strengths:**

*The paper exploits an interesting analogy between the current score matching framework used for generative modelling and SDEs coming from physics and the inverse problems associated to them. The idea is that the forward process do not need to be a diffusion and therefore we can learn the backward process for a given SDE to sample a trajectory backward. This is indeed an interesting analogy that can lead to several interesting applications in the future.

*The numerical examples are clear and nicely illustrated. The numerical examples show that the current framework works better than simply running the simulator backwards or simply approaching it by a neural network.

**Weaknesses:**

* In line 70 the paper states that it's goal is to sample from the posterior distribution $p(x_0 | x_M^*)$ where $p$ corresponds to the forward physical process. But, during the remainder of the paper, one of the proposed algorithms suggested (SMDP-ODE) can not be considered as a sampler from $p(x_0 | x_M^*)$, since it is a function of $x_M^*$. As shown by [1], what holds is that the ODE pushes forward the distribution of $p(x_M)$ to a distribution that converges (weakly) to $p(x_0)$ so it is not clear what is the actual point of starting the ODE from a given (fixed) $x_M^*$. In the conclusion the authors do touch on the point that the ODE variant is not a sampler but I feel that the presentation generates a lot of confusion for the reader during most parts of the paper.

* As I understand it, only the toy problem presents a multimodal posterior. One would expect this kind of technique to be particularly useful in settings where multimodality of the posterior is present, so I would expect the authors to focus more on those cases. This is also reflected by the metrics being used (comparisons to the "true" $x_0$, either $RMSE$ or $LPIPS$), which would arguably make less sense in the case of multimodality and even in the case where the mode of the posterior distribution does not match the $x_0$ that produced the fixed $x_M^*$.

* The paper do not present any comparison with other inverse problems solvers, focusing on score matching based approaches only. Even though this in comprehensible to a degree, I feel there should be at least one of the non-score matching based approaches for solving inverse problems.


[1] Yang Song, et al. "Score-Based Generative Modeling through Stochastic Differential Equations." International Conference on Learning Representations. 2021.

Remarks:
* The posterior distribution is clearly defined, as being the distribution given by
$\int p(x_{0:M-1}, x_M^*) dx_{1:M-1}$, therefore I find it strange to say "a" posterior distribution in line 136.

**Questions:**

* There is a practical difference between training the proposed model and the standard denoising diffusion model [2]. In [2] the losses are calculated by sampling from $p(x_t | x_0)$ which can be written as $\mu_t(x_0) + \epsilon$ where $\epsilon$ is some gaussian noise. In the context of the paper this is not always possible and one need to rely on a given set of paths. How does this impact the training? Is it possible to generate the same kind of training function that does not depend of unrolling the paths?

* The values in Table 1 are counter-intuitive as far as I'm concerned. Why do the methods '1-step' and 'SSM-VR' seem to achieve the best posterior metric Q for smaller datasets in both ODE and SDE?

* Are the situations where using an inverse step solver $P^{-1}$ instead of $P$ motivated only by the numerical cost of running $P$ ?




[2] Ho, J., Jain, A., & Abbeel, P. (2020). Denoising Diffusion Probabilistic Models. In Advances in Neural Information Processing Systems (pp. 6840–6851). Curran Associates, Inc..

---

> ### Author Rebuttal · Authors · 2023-08-10
>
> We thank the reviewer for their valuable review and helpful suggestions.
>
> - **SMDP-ODE and $p(\mathbf{x}_0|\mathbf{x}_t)$**: We updated section 2.3 to more clearly distinguish the different design choices for the ODE and SDE sampler. Since the inference of the ODE method is similar to the trajectories of the multi-step loss based on maximum likelihood training it can be regarded as a maximum likelihood solution.
> - **Multimodal posterior**: As correctly stated by the reviewer, our proposed method is capable of sampling from multimodal posteriors similar to diffusion models. We have tested this extensively for the 1D toy problem. In more challenging high-dimensional problems, it is often difficult to obtain a ground truth posterior to which we can compare the posterior we obtain by sampling from the SDE if there is multimodality. Additionally, even if the ground truth posterior is known, defining a good metric to compare both high-dimensional posteriors for simulation data is non-trivial. Nonetheless, the heat diffusion experiments make a step in that direction as in our experiments the information from higher frequencies is lost and must be generated by the network. As demonstrated by the low spectral error, our network correctly synthesizes structures on smaller scales that match the ground truth data distribution.
> - **Additional baselines**: We have included two additional baselines for the buoyancy-driven flow with obstacles experiment. The first baseline is based on classical optimization with differentiable physics and the L-BFGS algorithm [1] . The second baseline is a hybrid approach that combines a standard diffusion model with gradient based optimization [2]. Both of them represent very strong baselines and highlight the effectiveness of our method in challenging high-dimensional inverse problems.
> - **Standard denoising diffusion model training and relying on a given set of paths**: In principle it is not necessary to rely on a specific set of paths. In standard diffusion models it is easy to sample points at a specific time index from the subspace of paths that connect the data distribution at t=0 to the noise distribution at t=T due to the simplicity of the underlying SDE. In our context, just sampling points that are available from the dataset would correspond to this approach and resembles the 1-step loss. However, it is easy to generate the paths for standard diffusion models during training (the points can be sampled easily) and an infinite number can be considered during training in theory whereas it is not so easy in our scenario since the paths can neither be easily generated nor expanded. In that sense, the multi-step loss increases the number of points in the training, by generating additional possible paths based on individual points in the training dataset based on the physics simulator and current score model parameter. If we consider extremely large training dataset as done for standard diffusion models in computer vision, we expect that the 1-step training and multi-step loss would attain a similar performance. Such large high-quality and diverse datasets are however rarely available for more specific scientific applications.
> - **Counter-intuitive values in Table 1**: The training and evaluation of this task is somewhat noisy. The posterior metric is very sensitive to the score field at a specific region (where the paths from -1 or 1 merge or separate). Because of this, even though the training converges and losses decrease, it is possible that the metric Q is close to 0, because there is an imbalance between the predicted classes (in these cases the evaluation gives either paths for -1 or 1 and has a problem with the multimodality in this example). The multi-step loss is much less noisy than the other training methods here.
> - **Inverse step solver $\tilde{\mathcal{P}}^{-1}$**: For all experiments where the physics simulator is not learned (experiments 1-3) we use the existing implementation of the forward solver to derive the reverse solver by changing $\Delta t$ to $-\Delta t$. This is not motivated by the numerical cost. When pretraining the inverse solver in experiment 4, we specifically use a solver pretrained for the inverse problem.
>
> [1] Thuerey et al. "Physics-based deep learning" arXiv:2109.05237
>
> [2] Chung, Hyungjin, Jeongsol Kim, Michael T. Mccann, Marc L. Klasky, and Jong Chul Ye. "Diffusion posterior sampling for general noisy inverse problems." arXiv:2209.14687

---

> > ### Comment · Area_Chair_mbRR · 2023-08-17
> >
> > Dear reviewer,
> >
> > In order to ensure the quality of the overall evaluation, please acknowledge the authors response and indicate whether you want to update or keep your original evaluation. This paper is in the borderline range and it would be helpful to have your feedback to make an informed decision.
> >
> > Thanks again for your time and effort.
> > The AC.

---

### Official Review · Reviewer_og5h · 2023-07-07

**Soundness:** 4 excellent
**Presentation:** 4 excellent
**Contribution:** 2 fair
**Rating:** 4
**Confidence:** 2

**Summary:**

The authors propose diffusion-based inverse problem solvers involving the temporal evolution of physics systems. The method utilize a combination of score function and an inverse physics simulator, which corresponds to reverse of drift term in diffusion models, to moves the system’s state backward in time. They demonstrate the effectiveness of their method in a wide range of inverse physics problems.

**Strengths:**

The authors propose multi-step loss to capture long-range dependence in physics system, which has potential implication for general diffusion models.
The experiment are conducted extensively.

**Weaknesses:**

The method is modification of diffusion model with adoption of inverse physic simulator in order to address nonlinear drift term of the physics system. In this regard, its novelty is limited and its applicability may be restricted to certain scenarios.

Minor/errata

Hats are missing in equation (4)


**Questions:**

In the evaluation of this methods, is it acceptable to not compare it with conventional method such as finite element methods?

**Limitations:**

It is suggested to include the future direction of this work.

---

> ### Author Rebuttal · Authors · 2023-08-10
>
> We thank the reviewer for their valuable review and helpful suggestions.
>
> - **Conventional methods as baseline**: We evaluated an additional baseline which represents a more traditional method for solving inverse problems for the buoyancy-driven flow with obstacles experiment. This baseline is based on classical gradient-based optimization for differentiable physics and the L-BFGS algorithm, see [1]. Results can be seen in the pdf attached to the global response.
> - **Future directions**: We now included future directions and extensions of this work in our conclusion section. In particular, we are interested in enhancing the trajectories and obtaining solutions by including gradient-based optimization during inference.
>
> [1] Thuerey et al. "Physics-based deep learning" arXiv:2109.05237

---

> > ### Comment · Area_Chair_mbRR · 2023-08-17
> >
> > Dear reviewer,
> >
> > In order to ensure the quality of the overall evaluation, please acknowledge the authors response and indicate whether you want to update or keep your original evaluation. This paper is in the borderline range and it would be helpful to have your feedback to make an informed decision.
> >
> > Thanks again for your time and effort.
> > The AC.

---

> ### Comment · Reviewer_og5h · 2023-08-17
>
> I have carefully reviewed other reviewers’ comments and authors’ responses. Although I didn't recognized it earlier, this work potentially carries significant contribution, being the first diffusion-based approach to partial differential equations. I also acknowledge the authors' proactive efforts to address various concerns raised.
>
> However, an important factor influencing my assessment, which I hadn’t recognized thus didn’t express in my initial review, is my limited familiarity with physical systems. As a result, I find it difficult to raise my evaluation of this paper, but I lower my confidence score in order to ensure a fair decision process.
>
> As mentioned earlier, this work has potential impact for researchers working on diffusion models. If you introduced physical systems in more detail in revised version, it could greatly capture the interest of researchers in the diffusion model domain. I look forward to an updated version that provides additional insights. For example,
> - Why is the PDEs in experiments important? What implications for the community is to compute reverse simulation accurately?
> - How meaningful is the inclusion of additional stochastic terms in the heat equation? Is it common practice to introduce stochastic terms into PDEs?
> - Given that this work employs real physical time ‘t’, in contrast to diffusion generative models that use imaginary ‘t’ to smooth complex distributions, does the inclusion of ‘t’ as input in the score function hold significance? Consider a trajectory $x(t), t \in [0, T]$ of physical system. If we encounter another scenario with initial condition $x(s)$, is the corresponding trajectory $\{x(t-s)\}, {t\in[0,T-s]}$? Can we infer that $s(t,x)=s(t-s,x)$?

---

> > ### Author Response · Authors · 2023-08-20
> >
> > We thank the reviewer for their suggestions and acknowledging the paper's contributions. We believe that the additional questions asked by the reviewer are very insightful and we try to give brief answers here:
> >
> > 1. Aside from possible general improvements for diffusion-based generative modeling, where a more general class of drift functions (based on all kinds of PDEs) coupled with, e.g., the training setup of this paper could improve the performance, speed of inference and quality of conditional samples, there are many applications in the area of scientific machine learning. Including the physics simulator in the diffusion process enables a general framework to tailor generative modeling and conditional sampling to specific, challenging inverse physics problems, e.g., in astronomy, geophysics or climate science. This yields very effective models that can be used to obtain high-quality samples for inverse problems and uncertainty quantification. As these models include a simulator as an inductive bias, they are able to produce accurate and high-quality samples even in areas with limited training data.
> > 2. In our experiment with the heat equation, we included additional stochastic terms to easily embed the heat diffusion PDE in the mathematical framework of SDEs and diffusion models. An advantage of this is that it theoretically implies that the distribution of samples from inference matches the correct posterior. The weight of the stochastic term was very small and had little to almost no effect on the overall dynamics of the PDEs. To our knowledge, similar approaches for other PDEs in the context of uncertainty quantification and diffusion models have not been considered so far, but we believe that a thorough analysis of this would be of great interest to researchers and leave it as a subject to future work.
> > 3. This question proposes a very interesting experiment. We have thought about omitting the time as input to the score model, however in the experiments considered in this paper the distribution of states $p_t$ at a specific point in time $t$ was always different than the distribution of states at a different point in time (e.g. the smoothness of states from $p_t$ in the heat equation changes with $t$). In these cases, additional information about the time $t$ will help the model. On the other hand, omitting $t$ could potentially improve the generalization capabilities of the model. It could also be very useful in exactly the cases mentioned by the reviewer, where $p_t \approx p_{t'}$ but $t \neq t'$. We leave additional experiments and an extended analysis of this to future work.

---

### Official Review · Reviewer_riEs · 2023-07-13

**Soundness:** 3 good
**Presentation:** 2 fair
**Contribution:** 2 fair
**Rating:** 6
**Confidence:** 3

**Summary:**

This paper proposes a diffusion-based unrolled strategy for learning solve ordinary differential equations (stochastic or not). After introducing the problem at stake and proposing two training strategies for solving it (namely a 1-step loss approach and a multi-step approach), the authors draw a theoretical parallel with denoising score matching and with probability flow ODEs. More precisely, the authors claim an equivalence between training an architecture with the proposed 1-step loss and training the same architecture with a score matching objective; and the equivalence between minimizing the multi-step loss and maximizing a variational lower bound. Eventually, the authors investigate the performance of the proposed approach in 4 setups, including deterministic and stochastic problems, ranging from a simple 1D experiment to a navier-stokes simulation.

**Strengths:**

- The paper investigates an original idea proposing to link traditional 1-step or multi-step losses in trajectory estimation for dynamical systems with diffusion and score matching. To the best of my knowledge, this idea is new in the literature and makes a link between traditional training approaches and diffusion processes.
- The authors perform a large number of experiments to support their claims.

**Weaknesses:**

- Albeit well written, the article is difficult to follow due to not always well organised sections and very large amount of information.
- The chosen baselines do not seem very relevant, because not relying on standard losses for training denoising diffusion architectures for inverse problems.
- There is a potential problem with one of the theoretical results (Theorem 2.1)
- The literature review is not sufficient; while important and recent works are appropriately cited, there lacks references to physics informed diffusion approaches for inverse problems.

**Questions:**

The following points list my concerns / questions / suggestions for the authors.

**1. Main concerns**

**1.1 Difficulties to follow the article** My main concern is that the paper is very difficult to follow. While overall well written, I tend to lose track of what the authors are aiming to do. For instance, in "method overview", the problem is not clearly stated. In my view, equation (6) of the main is the model that the authors aim at solving everywhere, but it comes in the middle of a section explaining score matching. In fact, the problem setting description from the appendix is rather clear while the first paragraph of 2 is not clear. There is a numerical update rule, but where does it come from? What is P? What is x? How are they related? Maybe if (6) had come in the introduction, the "Problem formulation" from section 2 would have been clearer.

**1.2 Comparison with baselines** While the theoretical explanations and intuitions link quite clearly denoising score matching with the problem of interest, the case where a network $s_{\theta}(x, t)$ is trained as a denoiser with a time embedding in a diffusion fashion does not seem to be present in the experiments (in ISM and SSM, $s_\theta(x, t)$ are not trained as denoisers if I understand correctly; please correct me if I'm wrong). Moreover, these architectures do not scale to higher dimensional problems (see line 225). As such, I wonder whether these baselines provide fair comparisons. Given the embedding of $s_\theta(x, t)$, I would suggest to use a method relying on denoising diffusion such as [HJA20]. If not applicable, maybe Diffusion Schrodinger Bridge would be a strong baseline to compare to [1]. Furthermore, given the similarities between the problem the authors want to tackle and the one from [1], I think adding a brief explanation on how the problems / approaches differ might be welcome.

**1.3 PDE litterature** I have difficulties with the relations to other works. In my opinion, section 4 comes too late and should be inserted way earlier first - probably before section 2. More precisely: in "Learned corrections for numerical errors", a more detailed review of learning-based methods for solving PDEs / stochastic PDEs would be helpful.

**1.4 Lack of references to physics informed diffusion approaches** In general I believe that a large part of the imaging inverse problems literature is not mentioned. While this is not the core topic of the paper, it remains interesting in its own right since many papers have proposed methods for incorporating a measurement operator (or P(x) in the authors' words) within diffusion models, thus making the diffusion process "physics aware", which is precisely what the authors want to do here. A cornerstone reference, which the authors included, is [Chu+22]. However, the authors mention that this work performs uncertainty quantification, and state "either focus on the denoising objective common for tasks involving natural images, or the synthesis process of solutions does no directly consider the underlying physics.": I disagree with this, see e.g. Figure 4 of the paper. If this reference does not convince the authors, here are other references where underlying physics / acquisition procedures are taken into account in a diffusion process: [2, 3, 4, 5]. Note also that an extensive literature in the inverse imaging literature has focused on a similar approach to your multistep loss, via architectures known as unfolded architectures incorporating the physics model inside the architecture [6].

**1.5 Proof of Theorem 2.1** I wonder whether the proof of $\Rightarrow$ is correct. My concern is with the particular sentence: "let $\theta^*$ denote a minimizer such that $\mathcal{L}(\theta) \to 0$ as $\Delta t \to 0$. Note that at least one minimizer exists as we can choose $s_{\theta^*}(x, t) = \nabla_x \operatorname{log} p_t(x)$." While I agree that a minimizer to this convex functional exists regardless of the nature of  $s_\theta(x, t)$, I am not sure you can assume that the minimum value of the functional tends to 0 without any further assumption on the very nature of $s_\theta(x, t)$: take for instance a simplistic model that is not powerful enough to approximate $\nabla_x \operatorname{log} p_t(x)$...

**2. Additional painpoints**

**2.1 Title** IMO, there is a mismatch between the title and the article: the authors do not propose a denoising score matching method, but a radically different approach that they claim to be equivalent to score matching.

**2.2 Spectral loss** Why is a spectral loss necessary (line 241)? Aren't l2 (or l1, often more efficient) sufficient? Choosing a spectral loss seems unusual to me, maybe linking to some other papers using a similar loss would be welcome?

**2.3 Cumberstone notations** Notations are sometimes difficult to follow, maybe clarifying them would be useful. Some examples: around line 167, $p_1$ and $p_{-1}$ clash with $p_0$ and $p_T$. Around line 63, $\Delta t = t_j-t_k$, but then the authors use the convention $(t_m)_{0 \leq m \leq M}$. Why not replacing $j$ and $k$ with $m+1$ and $m$? etc...

**2.4** Shouldn't the term inside the norm in eq. (2) be updated $x_{m+1}-x_m - \Delta t \cdots$?

**2.5** line 110 of the supplementary, should (9) not be (8) instead?

**2.6** Eq. (38) in supplementary: is the brownian term not missing?

**2.7** Table 1 from supplementary: DilatedConv --> Dil-ResNet?

**2.8** Line 426 of supplementary: 100% of what?

**2.9** line 565 of supplementary: define the exact expression for $s_1$ and $s_2$.


**References:**

[1] De Bortoli, Valentin, James Thornton, Jeremy Heng, and Arnaud Doucet. "Diffusion Schrödinger bridge with applications to score-based generative modeling." Advances in Neural Information Processing Systems 34 (2021): 17695-17709.

[2] Chung, Hyungjin, Jeongsol Kim, Michael T. Mccann, Marc L. Klasky, and Jong Chul Ye. "Diffusion posterior sampling for general noisy inverse problems." arXiv preprint arXiv:2209.14687 (2022).

[3] Zhu, Yuanzhi, Kai Zhang, Jingyun Liang, Jiezhang Cao, Bihan Wen, Radu Timofte, and Luc Van Gool. "Denoising Diffusion Models for Plug-and-Play Image Restoration." In Proceedings of the IEEE/CVF Conference on Computer Vision and Pattern Recognition, pp. 1219-1229. 2023.

[4] Kawar, Bahjat, Michael Elad, Stefano Ermon, and Jiaming Song. "Denoising diffusion restoration models." Advances in Neural Information Processing Systems 35 (2022): 23593-23606.

[5] Rout, Litu, Negin Raoof, Giannis Daras, Constantine Caramanis, Alexandros G. Dimakis, and Sanjay Shakkottai. "Solving Linear Inverse Problems Provably via Posterior Sampling with Latent Diffusion Models." arXiv preprint arXiv:2307.00619 (2023).

[6] Adler, Jonas, and Ozan Öktem. "Learned primal-dual reconstruction." IEEE transactions on medical imaging 37, no. 6 (2018): 1322-1332.

**Limitations:**

Yes.

---

> ### Author Rebuttal · Authors · 2023-08-10
>
> We thank the reviewer for their valuable review and helpful suggestions.
>
> - **Structure of sections and extended literature review**: In line with feedback from other reviewers, we have moved the related work section to appear after the introduction. We also removed some of the experimental details and pushed them to the appendix. With the new space we expanded the method overview section to include more explanations, similar to the extended method overview in the appendix. We also already discuss the physical system as an SDE (equation 6) in the introduction now. We also expanded the discussion of additional learning-based methods for solving PDEs / stochastic PDEs and physics-informed diffusion approaches in the related work. We hope that this improves the structure of the paper and facilitates following the main goals of the article while only making minor changes to the content.
>
> - **Additional baselines and physics informed diffusion approaches**:  We have included additional baselines for the buoyancy-flow problem, which represents the most challenging task. In particular, we include direct numerical optimization with the BFGS method and the differentiable solver as well one of the recent physics informed diffusion approaches [2] mentioned by the reviewer. Our method compares very favorably to these baselines for challenging tasks, see the pdf of the global response. The solutions obtained by classical optimization with BFGS and the differentiable solver attain a good reconstruction MSE, although still outperformed by SMDP. However, as can be seen in the visualizations, the solutions are far away from more natural flows contained in the training dataset. In additional to that, SMDP provides a significant speedup ($\sim$ 100x) since it does not rely on expensive gradient computations during inference. Also, our approach obtains better performance than [2], Algorithm 1 for this task. We believe that there are several reasons why our method performs significantly better than [2] in this situation. The training dataset is quite small for standard diffusion models as it comprises only 250 simulations. When training a DDPM to generate flow fields for a specific point in time, there are only 250 samples, which might not be sufficient. Additionally, the computation of the forward simulator $\mathcal{A}$ by the differentiable solver is very slow (>60 minutes for inference of a single simulation) and gradients backpropagated through many simulation steps are not very helpful anymore for the optimization. If we run [2], Algorithm 1 on this task, the initial state is too noisy and gradients are extremely high or contain NaNs. We therefore sample the first 900 steps with the standard DDPM algorithm and switch to [2] after that. However, this could degrade the performance of the method.
>
> - For the toy problem, where we compare with ISM/SSM, we are mainly interested in methods that learn the score to a given arbitrary SDE. For the toy problem, it would be trivial to learn the mapping between the distribution of initial states at $t=0$ and the Gaussian distribution at $t=10$ with a standard diffusion model or the diffusion Schroedinger Bridge method, but that was not the main objective of this task.
>
> - **Proof of Theorem 2.1**: We included an additional assumption for this theorem that the hypothesis space corresponding to the model architecture $s_\theta(x,t)$ includes the correct score $\nabla_\mathbf{x} \log p_t(x)$. This should fix the mentioned issue.
>
> - **Spectral loss**: Using a spectral error for scientific data is not uncommon, see e.g. Um et al. (2020). The spectral error facilitates evaluating how the prediction matches the statistical properties of the ground truth on different scales, which is not possible with the l2 error.
>
> - **Notation issues**: We thank the reviewer for the detailed comments. We have fixed all mentioned issues in the main paper and supplementary where appropriate.
>
> We believe these updates address the issues raised by the reviewer, and hence we kindly ask the reviewer to consider raising their score.

---

> > ### Comment · Reviewer_riEs · 2023-08-11
> >
> > I thank the authors for their clarifications; I am looking forward to reading the updated version of the paper.
> >
> > **1.** I thank the authors for running experiments with [2]. I am not surprised that their method performs better due to both the small number of samples and the differentiability issues of $\mathcal{A}$.
> >
> > **2.** A major difference holds between the pointed references (including [2]) and the work of the authors, namely: in [2] and others, $s_\theta(x, t)$ is trained to remove Gaussian noise from images. It is not trained to deblur / restore / inpaint etc, in other words, $s_\theta(x, t)$ is not shown any measurement operator during the training procedure, which is a strong advantage of [2] ($s_\theta(x, t)$ does not need to be retrained for new imaging inverse problems, and might generalize to inverse problems where almost no training data is available, just like the case studied by the authors). I believe that it is important to stress this difference in the paper.
> >
> > **3.** I appreciate the effort to correct the assumptions of Theorem 2.1, but I am afraid that this is not sufficient since problem (4) is of course *not* convex in $\theta$ (unless very strong assumptions on $s_\theta$). Assume that $\nabla_x \log p_t$ belongs to the hypothesis space, there still exists local minimizers such that $\mathcal{L}(\theta) \nrightarrow 0$. I may very much be wrong, but I don't think the proof in its current form might be fixed as such. Instead, would it be possible to replace the $\mathcal{L}(\theta) \rightarrow 0$ by $\mathcal{L}(\theta) \rightarrow \varepsilon$ for some small $\varepsilon$? If this is not possible, I find that $\Leftarrow$ is a nice result that is sufficient in itself.
> >
> > **4.** I agree that 250 samples is too few to train a meaningful diffusion model. However, $s_\theta(x, t)$ would traditionally be trained as a pure denoiser, for which 250 samples is enough (with data-augmentation). Furthermore, since the proposed algorithm includes the physics of the problem, it is possible that this smaller, toy diffusion model might be sufficient. After all, the authors are not trying to generate high quality data from pure noise, but to revert a physical process starting with some data. Do the authors confirm that they have not tried to train $s_\theta(x, t)$ as a denoiser? (I am not expecting the authors to run this experiment, but it migth be of interest for future work.)

---

> > > ### Author Response · Authors · 2023-08-17
> > >
> > > We thank the reviewer for their response and suggestions.
> > >
> > > - We agree with the reviewer that there is a methodological difference between our work and the work by [2] and others, which we will stress in the paper. We have not tested using the trained $s_\theta(x,t)$ for other tasks, but because training includes the physics simulator, $s_\theta(x,t)$ might generalize less well to other problems than training based on a denoising objective. On the other hand, this enables our method to attain excellent performance on specific individual tasks even with little training data, which is difficult with more general approaches to inverse problems such as [2] and others, due to, e.g., the mentioned issues of the expensive measurement operator. We believe this represents a noteworthy empirical result.
> > >
> > > - We are not sure, if we understand the concern correctly. For "$\Rightarrow$", we assume that there is a sequence $\Delta t_1, \Delta t_2, ...$ with $\lim_{n \to \infty} \Delta t_n = 0$ and a sequence $\theta_1, \theta_2, ...$, where $\theta_n$ is a minimizer to the objective $L_\mathrm{single}^{\Delta t_n}(\theta)$ that depends on the step size $\Delta t_n$. If there is $\theta^*$ such that $s_{\theta^*}(x, t) \equiv \nabla_x \log p_t(x)$, then $L_\mathrm{single}^{\Delta t_n}(\theta_n) \leq L_\mathrm{single}^{\Delta t_n}(\theta^*)$. From "$\Leftarrow$" we know that $\lim_{n \to \infty} L_\mathrm{single}^{\Delta t_n}(\theta^*) = 0$ and therefore also $\lim_{n \to \infty} L_\mathrm{single}^{\Delta t_n}(\theta_n) = 0$. Note that we do not try to find one of possibly multiple global minima of $L_\mathrm{single}^{\Delta t_n}(\theta)$ here, but instead assume that $\theta_n$ is returned by some optimization process. As mentioned by the reviewer, the objective is not convex in most cases. It might also be the case that  $\theta_n \nrightarrow \theta^*$, but we are only interested in showing that $s_{\theta_n}(x,t) \to s_{\theta^*}(x,t)$.
> > > - We have trained $s_\theta(x,t)$ with a similar setup as denoisers and found that while training and inference work well, backpropagating gradients through multiple steps yields improved results for the buoyancy-driven flow experiment. To give more details about the exact setup: in appendix Figure 6, we compare training and inference with "joint" (method as explained in main paper) and "separate updates". The main difference between them is also explained in appendix Figure 1. For the 1-step training with separate updates, we draw samples $(x_i,t_i)$ and $(x_{i+1}, t_{i+1})$ from the dataset, predict $\hat{x_i}$ from $x_{i+1}$ using the simulator (physics step), add noise to $\hat{x_i}$ and denoise the state again with $s_\theta(x, t)$ (denoising step). Our evaluation for this experiment showed that while the 1-step training is simple to implement and very memory-efficient, the multi-step training obtains an improved performance for longer rollouts during inference.

---

> > > > ### Comment · Reviewer_riEs · 2023-08-17
> > > >
> > > > I thank the authors for their clarifications.
> > > >
> > > > - Regarding the training of $s_\theta(x, t)$ as a denoiser, I thank the reviewers for pointing my misunderstandings of their experiments. I believe that the experiment from section A.3 is very interesting and could be mentionned earlier in the paper - not only after section 3.3.
> > > >
> > > > - Why should one have $L_\text{single}^{\Delta t_n}(\theta_n) \leq L_\text{single}^{\Delta t_n}(\theta^*)$? If $\theta_n$ is a (local) minimizer to the (nonconvex) objective $L_\text{single}^{\Delta t_n}(\theta_n)$, we can only assume that $\nabla_{\theta} L_\text{single}^{\Delta t_n} (\theta_n) = 0$, but I fail to understand the argument. Could the authors please elaborate?

---

> > > > > ### Author Response · Authors · 2023-08-19
> > > > >
> > > > > We thank the reviewer for the helpful discussion and feedback.
> > > > >
> > > > > - We will update the paper to mention the experiment from appendix A.3 and E as well as its connection to the denoising objective earlier in section 3 as this experiment might be even more interesting to readers than we previously anticipated.
> > > > >
> > > > > - In "$\Rightarrow$" of the theorem, we assume that $\theta_n$ is the global minimizer of the objective $L_\text{single}^{\Delta t_n}(\theta)$. Therefore, for this direction the theorem states that $s_{\theta_n}(x,t) \to s_{\theta^*}$, where $\theta_n$ is the global minimizer to $L_\text{single}^{\Delta t_n}(\theta)$. Indeed, a similar statement about a local minimizer $\theta_n$ would be very difficult if not impossible, as this would strongly depend on the convexity of $L_\text{single}^{\Delta t_n}(\theta)$ and properties of $s_\theta(x,t)$. We will clarify this in the theorem and proof to avoid confusion and possible misunderstandings.
> > > > >
> > > > > We believe that we have addressed most of the reviewer's concerns. We therefore kindly ask the reviewer to update or reconfirm their score.

---

> > > > > > ### Comment · Reviewer_riEs · 2023-08-19
> > > > > >
> > > > > > I thank the authors for their clarifications. I was mislead by the formulation "a minimizer", which I implicitely understood as "a local minimizer", as opposed to "the minimizer", referring to "the global minimizer".
> > > > > >
> > > > > > Following discussions with the authors, I am willing to increase my score. My two main concerns, which were (i) the clarity of the paper and (ii) the proof of Theorem 2.1, have been resolved.

---

### Official Review · Reviewer_h6fX · 2023-07-25

**Soundness:** 2 fair
**Presentation:** 3 good
**Contribution:** 2 fair
**Rating:** 6
**Confidence:** 3

**Summary:**

This work describes a new method for solving inverse problems for time-evolved physical systems. It does this by combining two components: a time-independent reverse physics simulator (based on a priori domain knowledge or learned) and a learned time-dependent correction term. The posterior for the initial state can then be obtained by sampling from a related SDE (or a related ODE). This method is referred to as score matching via differentiable physics (SMDP). The authors argue that the learned correction term corresponds to the score (the gradient of the log-likelihood of the trajectory), providing a probabilistic interpretation of their model.

The method is numerically evaluated on four different physical systems, where it is shown to yield results superior to various baselines. This is the case for both the SDE and ODE variants of the method. The authors also discuss relations with existing methods for inference based on score matching as well as generative methods such as diffusion models.


**Strengths:**

The paper is well laid out, clearly explaining the task considered and the proposed solution. Various quantities and models are well defined and motivated. The numerical section is also well presented, with appropriate comparison to baseline methods such as implicit score matching and sliced score matching for one of the tasks and other neural network models for another.


**Weaknesses:**

There is little discussion of previous work on the inverse problem studied in this work. Indeed, it is only towards the end of the work that the authors discuss a set of related work, with little or no discussion of methods applied to the inverse problem in question. If it is the case that this particular problem has not been studied and therefore there are no state-of-the-art methods, this should be clearly spelled out.

Another issue is that the central theoretical result (that of the equivalence of the correction term to the score function of the system) is not empirically verified. This should be done since the theoretical result only holds in the continuous limit, so it is not clear how this applies in the discrete case. While this may not be possible in the more complex tasks studied, it should be doable for the 1D toy problem considered in Section 3.1. Due to the simplicity of the model, it should be possible to calculate the score function and compare it to the learned correction term.

Another issue is that the tasks are very close to the actual application, and a more detailed description may be necessary for readers who are not domain experts. For example, it is not clear what is meant by “We use semi-Lagrangian advection for the velocity and MacCormack advection for the hot marker density.”.

Overall, the novelty of the proposed method is not very high. This is a learned correction to a standard method of reverse time-stepping a simulation. Perhaps a more careful study of the learned correction term and its properties would help here, but as it stands, the results are incremental.


**Questions:**

– How is the problem in Section 2 motivated? Is this supposed to be a discretization of an SDE?

– Why is the step *added* to x_m in eq. (1) in order to obtain x_{m+1}? If this is stepping backwards, shouldn't it be subtracted? The conclusion on line 113 would make more sense in that case.

– In Section 2, N trajectories indexed by i from 1 to N are introduced, but never make another appearance. Instead, eq. (2) and (3) refer to an expectation over trajectories. How are we to think of the trajectories, as samples or as a random vector?

– Some more discussion as to the difference between the SDE and ODE variants would be useful. If I understand correctly, the former sampled the posterior states, while the latter provides an approximation of its mode (calculating the maximum a posteriori through a normalizing flow-type construction).

– Figure 2 is hard to parse. In (a), are we observing the true score or its approximation by the correction term? Also, it is not clear what is shown in (c), especially the trajectories at the bottom. Are these exploding trajectories occurring for GRID or for MLP? Are they for one step or multiple steps? The discussion of these results (lines 178–186) are similarly hard to follow.

– The two sentences on lines 212–215 seem to contradict one another (if the forward solver cannot be used, how are we implementing the reverse step using the forward solver?).

– Why are the results of the SDE not shown in Figure 5?

– Why are we adding Gaussian noise to each state in Section 3.3? If this is a deterministic system, should we not be considering deterministic methods to solve it?

– The conclusion on lines 302–303 contradicts the results in Figure 6(b) (ODE obtains better spectral error, not MSE).

– In various places, the authors use a period as a thousands separator. This should be a space in an English-language context.


**Limitations:**

The authors briefly discuss limitations (and possible future directions) in the last section. I don't believe there are any potential negative societal impact from the work that should be considered.

---

> ### Author Rebuttal · Authors · 2023-08-09
>
> We thank the reviewer for their valuable review and helpful suggestions.
>
> - **Previous work for inverse problem tasks**: We have updated the related work section of the paper to include additional references to prior work on inverse problems, particularly the heat equation. The buoyancy-driven flow task is new with no prior state-of-the-art methods, as we wanted to design a suitable problem that is high-dimensional and includes non-trivial physics simulators with difficult-to-learn dynamics (caused by placements of different obstacles). We have included additional baselines (see global response) for general inverse problem approaches to further highlight the effectiveness of our approach compared to state-of-the-art methods. As also suggested by other reviewers, we extended the related work and moved problem-specific details to the appendix. We restructured the paper by placing the related work section right after the introduction.
> - **Empirical verification of Theorem 1**: We have included new results in our global response where we analytically compute the correct score and compare it to the learned score of our methods (1-step, multi-step). This demonstrates that the 1-step training learns the correct score very accurately, whereas the multi-step also learns an approximation to the score, but overall weighs the prior implicit in the training dataset higher, which can be seen visually in the learned score field representation.
> - **Novelty of method**: We want to highlight that a diffusion-based approach to reverse a simulation in a probabilistic way has to the best of our knowledge not been considered in prior work. By theoretically linking learned correction training with the 1-step and multi-step loss to the score matching objective and maximum likelihood training, this approach can be very efficiently and reliably used for a large number of specific scientific downstream applications that rely on inverting highly non-linear dynamics and uncertainty quantification.
>
> - **Answers to questions**: While the training setup can be applied in more general settings (where the noise is not Gaussian), in our experiments and theory section, we consider Gaussian noise. In this case this is an SDE with a step size $\Delta t$ that depends on specific applications. Trajectories should be thought of as vectors (the size of the vector grows with the sliding window). We will expand the discussion of differences between ODE and SDE inference in the main paper. Your understanding is correct that the ODE inference generates a maximum likelihood solution whereas the SDE samples from the entire posterior. Fig. 2(a) shows the approximation of the score. We have included a comparison of the actual (analytic) score of an SDE and the learned correction in the global response pdf. Exploding trajectories in (c) occur in the GRID model trained with 1-step loss. For GRID multi-step training, there are no exploding trajectories. In line 212-215, we mean that we cannot use a single step backward because of exploding pixel values, but instead small steps + corrections are possible. In Fig. 5 SDE results are not shown because of space limitations, but they are shown in the supplementary material (Figure 8). For deterministic systems, there is often still noise in the measurement and another practical reason is that adding small noise helps performance in the case of multimodal posteriors. There was an error in Fig. 6b where the colors of ODE and SDE were swapped. The sentences in the text are correct, we apologize for the confusion.

---

> > ### Comment · Area_Chair_mbRR · 2023-08-17
> >
> > Dear reviewer,
> >
> > In order to ensure the quality of the overall evaluation, please acknowledge the authors response and indicate whether you want to update or keep your original evaluation. This paper is in the borderline range and it would be helpful to have your feedback to make an informed decision.
> >
> > Thanks again for your time and effort.
> > The AC.

---

> ### Comment · Reviewer_h6fX · 2023-08-18
>
> I would like to thank their authors for their response to my review and those of the other reviewers. In light of the changes made the the manuscript, I have decided to increase my rating.

---

### Official Review · Reviewer_SQFk · 2023-07-28

**Soundness:** 3 good
**Presentation:** 2 fair
**Contribution:** 2 fair
**Rating:** 6
**Confidence:** 2

**Summary:**

The paper proposes an approach to sample from the posterior distribution based on score-based diffusion models with a particular focus on inverse problems in physics.

The proposed approach, to my understanding, has two novel contributions:
- they use reverse-physics simulations and augment them with score estimates that supposedly leads to superior performance in sampling,
- they propose a multi-step training regime where scores are estimated in sequential manner for a given time-horizon, in contrary to the standard single-step score estimation.

The method was tested on different synthetic data experiments, and both the proposed approaches seem to produce meaningful improvements.

**Strengths:**

- The multi-step training of the score function seems to be effective in synthetic data applications.
- The theoretical results, although not particularly novel, provide a complete picture of the proposed methods.
- The experiments presented validate the proposed ideas well.

**Weaknesses:**

- One of the novelties claimed in the paper is the use of score-function to simply _refine_ the outputs of a reverse-physics simulator. Can't one simply view the reverse-physics simulator as (non-learned) _part of_ the parametric score model? In this case, the claim simply becomes that a physics-informed model to approximate the score is better than one that is oblivious to the physics? Isn't this an unsurprising statement? This is the underlying motivation behind physics-informed neural nets (PINNs), a relatively large area of research. Could the authors clarify?
- In some of the experiments, the authors use the LPIPS metric to evaluate the quality of the solution. This does not make much sense as LPIPS is designed to evaluate the "perceptual" quality of the image, which has nothing to do with the physical accuracy (the metric one cares about).

**Questions:**

- How is the ground truth obtained in the case of SDEs?
- In Section 3.4 (Fig. 6), why is the spectral error worse for SDEs compared to ODEs while it is the other way around in other experiments?

**Limitations:**

- Releasing the code could be crucial as there are many delicate details one might need to get right for the method to work.

---

> ### Author Rebuttal · Authors · 2023-08-10
>
> We thank the reviewer for their valuable review and helpful suggestions.
>
> - **Refinement of model outputs**: We agree that the reverse-physics simulator and the score network can be unified in a single joint model. Nonetheless, a clear distinction between the simulator and score network makes sense from a methodological viewpoint, as the learned score is data-driven with little to no inductive bias at all, whereas the simulator includes only inductive biases and conservation laws. Also, during inference, the update step by the simulator can be regarded as a step between different points in time, whereas the score network corrects the simulation state by projecting it towards a region with higher likelihood at the same point in time. One can expect an improved performance from a physics-aware score-matching approach, but nonetheless this connection has not been made before in the form we present it. As such, we believe our results and theory provide an important basis for future work at the intersection of both methods.
> - **LPIPS metric**:  Our primary metric for simulation data is the L2 distance. However, the L2 distance can become very large when the prediction does not match the ground truth on a per pixel basis, even though the prediction might still be close to the ground truth visually. In these situations the LPIPS distance can be more informative as it is less sensitive to the exact pixel positions.
> - **Ground truth for SDEs**: We draw individual samples from the testing dataset at time 0 and simulate them forward in time. The prediction is compared to this sample via the spectral error and reconstruction MSE, which are both less sensitive to the exact pixel values of the sample.
> - **Spectral error in Fig. 6**: In this specific plot, there is a mistake and the colors for the SDE and ODE should be switched. We noticed this mistake shortly after submission but could not fix it any more. We apologize for the confusion.

---

> > ### Comment · Area_Chair_mbRR · 2023-08-17
> >
> > Dear reviewer,
> >
> > In order to ensure the quality of the overall evaluation, please acknowledge the authors response and indicate whether you want to update or keep your original evaluation. This paper is in the borderline range and it would be helpful to have your feedback to make an informed decision.
> >
> > Thanks again for your time and effort.
> > The AC.

---

### Author Rebuttal · Authors · 2023-08-10

We thank all reviewers for their helpful suggestions and comments. Based on the feedback, there are several updates that are of interest to all reviewers:

- **Additional baselines and physics informed diffusion approaches**:  We have included additional baselines for the buoyancy-driven flow problem, which represents the most challenging task. In particular, we include direct numerical optimization with the BFGS method and the differentiable solver as well as a recent physics informed diffusion approach [1] mentioned by one of the reviewers. Our method compares very favorably to these baselines for challenging tasks while inference is significantly faster at the same time, see the attached pdf.

- **Test-time distribution shifts**: We have tested the effects of test-time distribution shifts for the heat equation experiment. Here we train the score network for a specific combination of diffusivity and noise and vary both parameters for testing (always updating both the simulator and test ground truth), see the attached pdf. Overall, for small changes of the parameters (<30%), there seems to be very little overfitting. Changes in the reconstruction MSE and spectral error can mainly be explained by variations of the parameters making the task itself easier/harder to which our network generalizes nicely (e.g. less noise or higher diffusivity -> smaller reconstruction error).

- **Empirical verification of Theorem 1**: We have included new results in the attached pdf where we analytically compute the correct score and compare it to the learned score of our methods (1-step, multi-step). This demonstrates that the 1-step training learns the correct score very accurately, whereas the multi-step also learns an approximation to the score, but overall pulls trajectories during inference closer to the training data set distribution, which can be seen visually in the learned score field representation.

- **Structure of sections and extended literature review**: In line with feedback from other reviewers, we have moved the related work section to appear after the introduction. We also removed some of the experimental details and pushed them to the appendix. With the new space we expanded the method overview section to include more explanations, similar to the extended method overview in the appendix. We also already discuss the physical system as an SDE (equation 6) in the introduction now. Moreover, we expanded the discussion of additional learning-based methods for solving PDEs / stochastic PDEs and physics-informed diffusion approaches in the related work. We believe this improves the structure of the paper and facilitates following the main goals of the article while only making minor changes to the content.

[1] Chung, Hyungjin, Jeongsol Kim, Michael T. Mccann, Marc L. Klasky, and Jong Chul Ye. "Diffusion posterior sampling for general noisy inverse problems." arXiv preprint arXiv:2209.14687

---

### Comment · Area_Chair_mbRR · 2023-08-15
**Author-reviewer discussion**

Dear all,

The author-reviewer discussion period has now started. It will continue for one more week, until August 21.

@authors: Please respond to the comments or questions reviewers may further have. Remain short and to the point.

@reviewers: Please read the author's responses and ask any further questions you may have. To facilitate the decision by the end of the process, please also acknowledge that you have read the responses and indicate whether you want to update your evaluation.

- You can update your evaluation positively (if you are satisfied with the responses) or negatively (if you are not satisfied with the responses or share other reviewers' concerns). Please note that major changes are a reason for rejection.
- You can also keep your evaluation unchanged. In this case, please indicate that you have read the responses and that you do not have any further comments.

Best regards,
The AC

---

### Decision · Program_Chairs · 2023-09-21

**Decision:**

Accept (poster)

**Comment:**

The paper has received mixed reviews (6-6-6-4-4-7). The reviewers have raised a number of concerns, many of which have been addressed by the authors during the author-reviewer discussion period. In particular, reviewer mtJA's concerns (borderline reject) about the presentation and the consolidation of the empirical evaluation have been, in my opinion, satisfactorily addressed. Reviewer og5h's concerns (borderline reject) regarding the novelty have been cleared out during the discussion, although the evaluation has remained the same due to a lack of familiarity with the topic. Overall, I believe the reasons to accept the paper outweigh the reasons to reject it. The authors are requested to address any remaining concerns in the final version of the paper.